# Principal neuron diversity in the murine lateral superior olive supports multiple sound localization strategies and segregation of information in higher processing centers

Hariprakash Haragopal[1] & Bradley D. Winters [1,2 ✉]

Principal neurons (PNs) of the lateral superior olive nucleus (LSO) in the brainstem of mammals compare information between the two ears and enable sound localization on the horizontal plane. The classical view of the LSO is that it extracts ongoing interaural level differences (ILDs). Although it has been known for some time that LSO PNs have intrinsic relative timing sensitivity, recent reports further challenge conventional thinking, suggesting the major function of the LSO is detection of interaural time differences (ITDs). LSO PNs include inhibitory (glycinergic) and excitatory (glutamatergic) neurons which differ in their projection patterns to higher processing centers. Despite these distinctions, intrinsic property differences between LSO PN types have not been explored. The intrinsic cellular properties of LSO PNs are fundamental to how they process and encode information, and ILD/ITD extraction places disparate demands on neuronal properties. Here we examine the ex vivo electrophysiology and cell morphology of inhibitory and excitatory LSO PNs in mice. Although overlapping, properties of inhibitory LSO PNs favor time coding functions while those of excitatory LSO PNs favor integrative level coding. Inhibitory and excitatory LSO PNs exhibit different activation thresholds, potentially providing further means to segregate information in higher processing centers. Near activation threshold, which may be physiologically similar to the sensitive transition point in sound source location for LSO, all LSO PNs exhibit single-spike onset responses that can provide optimal time encoding ability. As stimulus intensity increases, LSO PN firing patterns diverge into onset-burst cells, which can continue to encode timing effectively regardless of stimulus duration, and multi-spiking cells, which can provide robust individually integrable level information. This bimodal response pattern may produce a multi-functional LSO which can encode timing with maximum sensitivity and respond effectively to a wide range of sound durations and relative levels.

[1] Department of Anatomy and Neurobiology and Hearing Research Group, Northeast Ohio Medical University, Rootstown, OH, USA. [2] Brain Health Research Institute, Kent State University, Kent, OH, USA. ✉email: bwinters@neomed.edu

The ability to localize sounds is critical for survival. Unlike vision, the auditory system has no intrinsic representation of space and sound source location information must be computed centrally. Two important cues used for horizontal/azimuth sound localization are interaural time and level differences (ITD, ILD). This information is extracted by neuronal circuits in the brainstem that compare information from the two ears[1–3]. Principal neurons (PNs) of the lateral superior olive (LSO) perform a subtractive analysis of excitatory inputs driven by the ipsilateral ear and inhibitory inputs driven by the contralateral ear and project to higher auditory processing centers in the inferior colliculus (IC) and nuclei of the lateral lemniscus. The primary function of this circuit has long been thought to be extraction of ILDs[4–6], however, it is increasingly appreciated that LSO neurons are also sensitive to ITDs for onsets, amplitude modulated sounds, and transient broadband sounds, such as rustling, that are common in nature and have high ecological importance[7–9]. These findings and inferences based on the evolution of giant synapses (calyx of Held) and other features in the contralateral pathway that deliver sound information with great speed and temporal precision has even led to the proposal that encoding relative timing is the primary sound localization function of the LSO[7,10].

Extraction of sensory information by the central nervous system often utilizes cellular properties tuned for specific functions[11] and ILD/ITD extraction places disparate demands on LSO neurons. LSO PNs are not a uniform group and there are aspects of cellular diversity in the LSO that are not fully explored. Thus, understanding cellular diversity in the LSO may provide insights into its function. A major gap in our knowledge is that intrinsic differences between inhibitory (glycinergic) and excitatory (glutamatergic) LSO PNs have not been assessed. Here we examine their morphology and intrinsic electrophysiological properties with respect to tonotopic location and firing type.

## Results

### Reporter mice can be used to efficiently differentiate inhibitory and excitatory LSO PNs.
LSO PNs are thought to be glycinergic and glutamatergic[12–16]. Sodium- and chloride-dependent glycine transporter 2 (GlyT2, Slc6a5) is required to bias vesicle filling to glycine and has been used to identify glycinergic cells in mice[17,18]. Vesicular glutamate transporter 2 (vGlut2, Slc17a6) is the dominant vGlut in the LSO[19] and appears to be obligatory in glutamatergic LSO neurons[16]. Olivocochlear neurons also found in the LSO of rodents are not glutamatergic or glycinergic but utilize a variety of other transmitter systems including GABA, dopamine, and acetylcholine[20–24] and were distinguished from PNs by morphology and electrophysiological traits (see "Methods"). Using in situ hybridization, it was found that GlyT2- and vGlut2-expressing cells do not overlap in the gerbil LSO[25] and in rats vGlut1/2 expression did not overlap with the vesicular inhibitory amino acid transporter required for glycine/GABA neurotransmission[16]. These findings suggest that inhibitory and excitatory LSO PNs are distinct cell types. To distinguish inhibitory and excitatory LSO PNs, we used a knock-in vGlut2 reporter mouse (see "Methods") in which endogenous vGlut2 expression is not perturbed and separate, soluble fluorophore (tdTomato) is produced[26–28]. We selected putative excitatory LSO PNs by their prominent somatic red fluorescence (Fig. 1a i, ii). Other LSO PNs exhibited a distinct lack of fluorescent labeling in their soma rendering these putative inhibitory cells "black holes" (Fig. 1a iii, iv). We also found that cells with high glycine immunoreactivity did not exhibit vGlut2-driven tdTomato expression (Fig. S1). Cells were filled with fluorescent dye (Fig. 1b) and roughly placed within the LSO (Fig. 1c).

We made whole-cell patch-clamp recordings from 48 vGlut2 positive (excitatory) and 41 vGlut2 negative (inhibitory) LSO PNs from mice at postnatal day 21–44 (P27 ± 0.59 avg). For comparison to recent in vivo reports on the function of LSO PNs (see "Discussion"), we also made current-clamp recordings from 39 unidentified LSO PNs from Mongolian gerbils (P21-36, 28 ± 0.67 avg). We present these data for qualitative evaluation and interpretation of findings but did not perform statistical comparison with mice for lack of equivalent groupings.

### Onset-burst and multi-spiking LSO PNs are found in both transmitter types.
We observed two action potential (AP) firing types in mouse LSO PNs at higher current injection levels; there were semi-phasic responses, which we refer to as onset-burst, in which cells fired 2–5 APs within 8 ms of stimulus onset (Fig. 1d i, iii) and multi-spiking neurons that fired throughout the stimulus duration at 200–550 Hz (Fig. 1d ii, iv). Two LSO PNs that fired single action potentials followed by small spikes/spikelets that did not meet our amplitude criteria for APs were included with onset-burst group because of their very similar response pattern. At lower current injection levels near rheobase, virtually all LSO PNs exhibited pure onset responses with single spikes. The only exception was one mouse cell that at rheobase exhibited an onset spike, but also a second stray spike near the end of the 200 ms pulse. In LSO PNs from gerbils, we also observed onset-burst cells (18/39, 46%) and multi-spiking cells (9/39, 23%) as well as truly phasic firing cells (12/39, 31%) that only fire a single AP regardless of current level.

While both firing types were observed in inhibitory/excitatory groups, inhibitory LSO PNs were more often onset-burst (31/41, 76%) while excitatory LSO PNs were evenly split between firing types (24/48, 50%). Excitatory multi-spiking LSO PNs exhibited somewhat greater variability in multi-spiking responses with 7 out of 24 (29%) having some irregular firing responses at intermediate current injection levels with pauses after initial firing or irregular gaps in firing. Inhibitory multi-spiking LSO PNs, while fewer, had more consistently sustained regular spiking responses (9/10). We found that some LSO PNs (9/89, 10%) changed their high-current-firing-type from initial characterization, just after break-in, to ~10 min later after cell dialysis when cells were categorized, and intrinsic properties measured. There were 5 excitatory and 1 inhibitory LSO PNs that changed from multi-spiking to onset-burst and 2 excitatory and 1 inhibitory LSO PNs that changed from onset-burst to multi-spiking. Additionally, most multi-spiking LSO PNs reduced their firing rate at a given current level over this time period (27/31, 51.2 ± 5% average), however, some increased (4/31, 128.0 ± 95.6%).

### Intrinsic membrane properties differ between LSO PN transmitter types.
In mice, we found that inhibitory LSO PNs had on average 4.9 mV lower resting membrane potentials ($p < 0.0001$, $t$-test, Fig. 1e left) and exhibited a wider range of resting membrane potentials. For consistent comparison of intrinsic membrane properties, cells were maintained at −66 mV with small current injections when necessary (67% of cells, I: 43.24 ± 25.31 pA, E: −59.73 ± 23.79 pA). Inhibitory LSO PNs had substantially higher input resistance (61% difference, $p < 0.0001$, $t$-test, Fig. 1f left) and correspondingly slower membrane time constants (34% difference, $p = 0.0004$, $t$-test, Fig. 1g left). Gerbils exhibited lower input resistances overall and especially low values not seen in mice. In mice, excitatory LSO PNs exhibited larger sag potentials in response to hyperpolarizing current injections ($p < 0.0001$, $t$-test, Fig. 1h left) suggesting higher hyperpolarization-activated cyclic nucleotide-gated (HCN) channel density which may contribute to

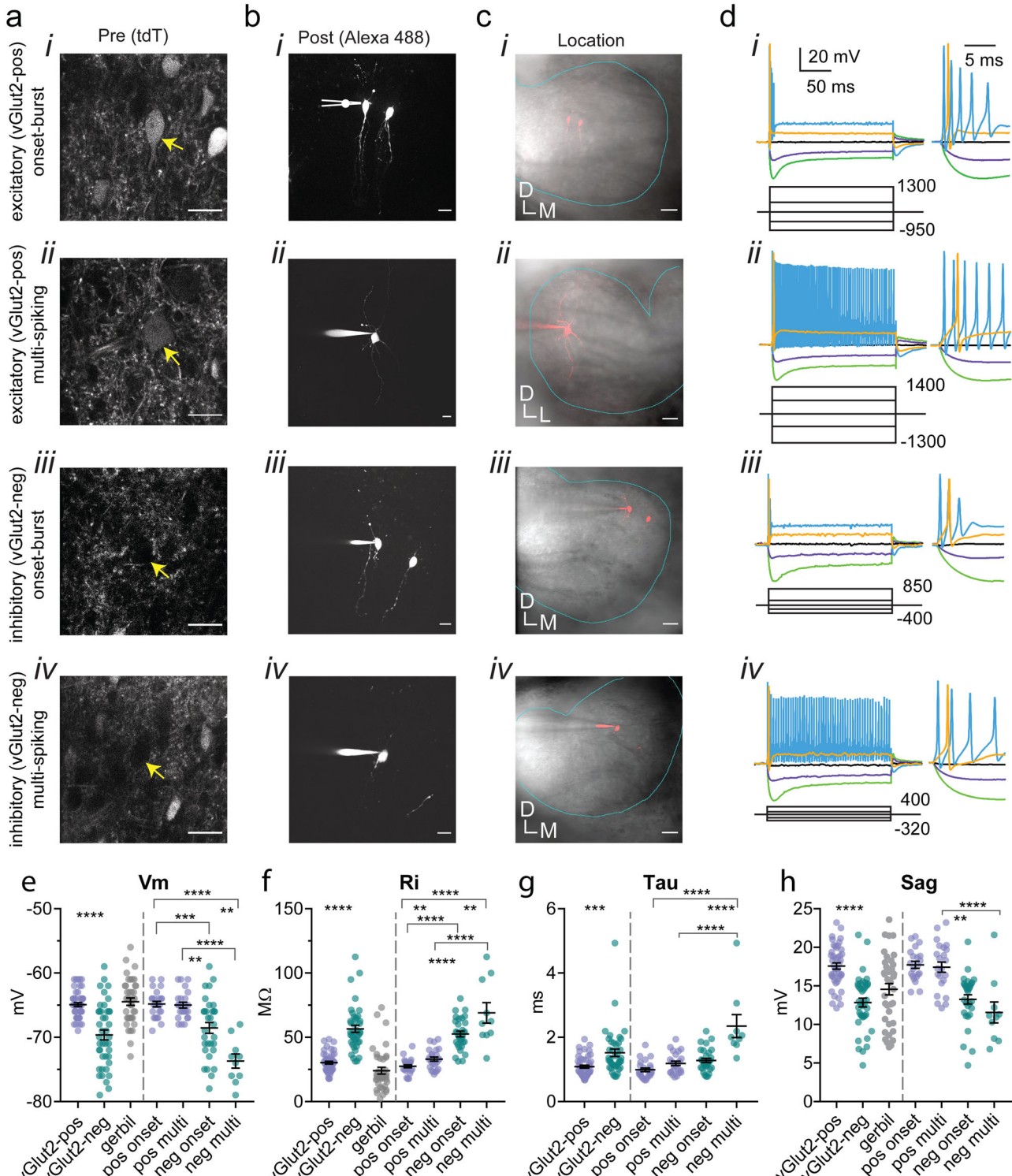

**Fig. 1 Inhibitory and excitatory LSO neurons both exhibit distinct onset-burst and multi-spiking firing types but have different intrinsic membrane properties. a** i–iv Example images of tdTomato (tdT) fluorescence prior to patch-clamping vGlut2-positive (excitatory, i, ii) and vGlut2-negative (inhibitory, iii, iv) LSO PNs (scale 20 μm). **b** i–iv Alexa 488 fluorescence post recording showing dendritic arborization. In some cases, previously recorded cells are also visible (scale 20 μm). **c** i–iv Combined Dodt contrast and fluorescence images showing location of recorded cells (red) relative to LSO boundary (cyan, scale 50 μm, D dorsal, M medial, L lateral). **d** i–iv Responses to current steps (below, pA) at the levels shown and responses at expanded time scale to the right. **e** Resting membrane potential. $n=$ cells(animals) from left to right: 48(30), 41(24), 39(23), 24(19), 24(19), 31(21), 10(9). **f** Input resistance. $n=$ cells(animals) from left to right: 48(30), 41(24), 39(23), 24(19), 24(19), 31(21), 10(9). **g** Membrane time constants obtained from small fixed current steps that produced depolarization of 1–1.5 mV. Equivalent data were not collected for gerbils. $n=$ cells(animals) from left to right: 47(30), 40(24), 24(19), 23(19), 31(21), 9(9). **h** Sag potentials at current steps with a hyperpolarization trough of ∼−96 mV. $n=$ cells(animals) from left to right: 44(28), 40(24), 39(23), 21(16), 23(18), 30(21), 10(9). Mean ± SEM. For pooled transmitter type data (left side bars) unpaired two-tailed $t$-test is shown. For firing type separated groups (right side bars), Tukey corrected multiple comparisons of two-way ANOVA is shown. $*p < 0.05$, $**p < 0.01$, $***p < 0.001$, $****p < 0.0001$.

their lower input resistance since HCN channels in the auditory brainstem are often partially active at resting membrane potentials[29–32]. Rebound spiking was observed in 39% of mouse LSO PNs and, in line with sag data, was more common in excitatory cells (I: 9/41, E: 26/48). We also observed some LSO PNs with AP afterhyperpolarizations (AHPs) with multiple components in both excitatory and inhibitory groups that was not associated with firing type in excitatory LSO PNs but was more common with inhibitory multi-spiking cells (excitatory onset-burst 5/24, 21%; excitatory multi-spiking 5/24, 21%; inhibitory onset burst 9/31, 29%; inhibitory multi-spiking cells 5/10, 50%). Sag amplitudes were different between the cells that had complex AHP and those that did not (complex: $13.75 \pm 0.90$ mV, single: $15.87 \pm 0.47$ mV, $p = 0.029$, t-test).

While our initial goal was to understand transmitter type differences, since we observed two firing types in both transmitter groups, we also tested whether and to what degree intrinsic membrane properties depended on transmitter or firing type using two-way ANOVA. For resting membrane potential, there were main effects of transmitter type ($F(1, 85) = 61.69$, $p < 0.0001$, Fig. 1e right) and firing type ($F(1, 85) = 11.32$, $p = 0.0012$), however, there was a significant interaction of firing and transmitter type ($F(1, 85) = 9.945$, $p = 0.0022$). The interaction accounted for 6.72% of the total variance while transmitter and firing type accounted for 41.7% and 7.65% respectively. For input resistance there were main effects of transmitter ($F(1, 85) = 118.3$, $p < 0.0001$, Fig. 1f right) and firing type ($F(1, 85) = 15.37$, $p = 0.0002$) and no interaction ($F(1, 85) = 3.758$, $p = 0.0559$). Transmitter accounted for 57.34% of the total variance while firing type accounted for 7.45%. Input resistance in gerbils also varied between firing types with onset-burst cells having values in between phasic and multi-spiking cells ($F(2,36) = [10.81]$, phasic: $11.94 \pm 1.83$ MΩ, onset-burst: $24.63 \pm 3.43$ MΩ, multi-spiking: $38.92 \pm 5.53$ MΩ, phasic vs. Onset-burst $p = 0.0363$, phasic vs. multi-spiking $p = 0.0001$, onset-burst vs. multi-spiking $p = 0.0308$, one-way ANOVA). For membrane time constant, in mice there were main effects of transmitter ($F(1, 83) = 45.10$, $p < 0.0001$, Fig. 1g right) and firing type ($F(1, 83) = 34.41$, $p < 0.0001$), however, there was a significant interaction of firing and transmitter type ($F(1, 83) = 16.25$, $p = 0.0001$). The interaction accounted for 11.27% of the total variance while transmitter and firing type accounted for 31.28% and 23.87% respectively. For sag potential, in mice there was a main effect of transmitter ($F(1, 83) = 15.78$, $p = 0.0002$, Fig. 1h right), but not firing type ($F(1, 83) = 0.0100$, $p = 0.9204$) and there was no interaction ($F(1, 83) = 3.072$, $p = 0.0833$). Collectively, these data suggest that transmitter type was the dominant factor for basic intrinsic membrane properties.

Consistent with their higher input resistances, in mice inhibitory LSO PNs had lower rheobase/activation threshold (40% difference, $p < 0.0001$, t-test, Fig. 2b left). Gerbils had overall higher rheobase and included cells with very high values not seen in mice, in excess of 1 nA, but were estimated from coarser current step sizes (~100 pA). Inhibitory LSO PNs in mice also had higher spike counts per given current level except at higher current levels in the onset-burst group (Onset-burst: $F(1, 1551) = 0.026$, $p = 0.87$, linear mixed model, post hoc F-test, Fig. 2c; Multi-spiking: $F(1,1551) = 46.22$, $p < 0.0001$, linear mixed model, post hoc F-test, Fig. 2d). Because inhibitory LSO PNs were more hyperpolarized at rest, we analyzed some excitability parameters at resting membrane potential as well. At resting membrane potential, input resistance differences were similarly large (68% difference, I: $61.91 \pm 3.42$ MΩ, E: $30.51 \pm 1.03$ MΩ, $p < 0.0001$, t-test). Using a coarser measure (100–150 pA steps) of rheobase at resting membrane potential in mice, we found that the difference between inhibitory/excitatory LSO PNs was still

large (23% difference, I: $380.26 \pm 21.86$ pA, E: $479.68 \pm 21.84$ pA, $p = 0.002$, t-test). For inhibitory LSO PNs a ~4 mV depolarization decreased rheobase by 52 pA ($p = 0.0002$, t-test) while in excitatory LSO PNs a ~1 mV hyperpolarization increased rheobase by 13 pA ($p = 0.096$, t-test).

AP threshold in mice was not different between transmitter groups ($p = 0.389$, t-test, Fig. 2e left) suggesting similar voltage gated sodium channel (VGSC) types although other factors such as hillock location or potassium channel interactions could impact this parameter. Inhibitory LSO PNs had lower AP peak voltages (16% difference, $p < 0.0001$, t-test, Fig. 2i left) and smaller AP peak amplitudes (20% difference, $p < 0.0001$, t-test, Fig. 2h left) suggesting lower density of VGSCs. Such differences in VGSC populations may affect how LSO PNs integrate sound-driven synaptic information.

AP half-width was wider in inhibitory LSO PNs in mice despite shorter APs ($p < 0.0001$, t-test, Fig. 2j left) suggesting high-voltage potassium channel differences. The AP rise slope was correspondingly slower in inhibitory LSO PNs ($p < 0.0001$, t-test, Fig. 2k left) as was the AP fall slope ($p < 0.0001$, t-test, Fig. 2l left).

For AP parameters, we also asked what the relative effects of transmitter and firing type were in mice using two-way ANOVA. For rheobase, there were main effects of transmitter ($F(1, 85) = 39.48$, $p < 0.0001$, Fig. 2b right) and firing type ($F(1,85) = 23.39$, $p = 0.0001$) with no interaction ($F(1, 85) = 0.2764$, $p = 0.6004$). Transmitter accounted for 28.72% of the total variance while firing type accounted for 17.01%. For threshold, there were no main effects or interactions (transmitter: $F(1, 85) = 1.31$, $p = 0.2556$, Fig. 2e right; firing: $F(1, 85) = 0.2649$, $p = 0.6081$; interaction: $F(1, 85) = 0.7702$, $p = 0.3826$) nor were there for threshold amplitude (transmitter: $F(1, 85) = 0.1937$, $p = 0.6610$, Fig. 2f right; firing: $F(1, 85) = 0.5409$, $p = 0.4641$; interaction: $F(1, 85) = 1.035$, $p = 0.3119$).

For AHP amplitude, there was a main effect of firing type ($F(1,85) = 15.27$, $p = 0.0002$, Fig. 2g right), but not transmitter ($F(1,85) = 0.4524$, $p = 0.5030$) with no interaction ($F(1,85) = 3.439$, $p = 0.0672$) suggesting that AHP is one AP property that is firing type dependent, however, post hoc analysis suggested that firing type differences were driven by the inhibitory group (Fig. 2g right), as there was no difference between the excitatory onset-burst and multi-spiking groups.

For AP peak amplitude, there were main effects of transmitter ($F(1,85) = 12.45$, $p = 0.0007$, Fig. 2h right) and firing type ($F(1,85) = 8.068$, $p = 0.0056$), however, there was a significant interaction of firing and transmitter type ($F(1,85) = 5.307$, $p = 0.0237$). The interaction accounted for 4.28% of the total variance while transmitter and firing type accounted for 10.04% and 6.51% respectively. For AP peak voltage, there were main effects of transmitter ($F(1,85) = 32.81$, $p < 0.0001$, Fig. 2i right) and firing type ($F(1,85) = 12.43$, $p = 0.0007$), however, there was a significant interaction of firing and transmitter type ($F(1,85) = 6.306$, $p = 0.0139$). The interaction accounted for 3.86% of the total variance while transmitter and firing type accounted for 20.06% and 7.60% respectively. In both AP amplitude and voltage post hoc analysis suggested that firing type differences were driven by the inhibitory group (Fig. 2h, i right).

For AP half-width, there were main effects of transmitter ($F(1,85) = 7.485$, $p = 0.0076$, Fig. 2j right) and firing type ($F(1,85) = 11.83$, $p = 0.0009$), however, there was a significant interaction of firing and transmitter type ($F(1,85) = 7.43$, $p = 0.0078$). The interaction accounted for 6.04% of the total variance while transmitter and firing type accounted for 6.09% and 9.62% respectively. For max AP rise, there were main effects of transmitter ($F(1,85) = 16.94$, $p < 0.0001$, Fig. 2k right) and firing type ($F(1,85) = 9.527$, $p = 0.0027$), however, there was a

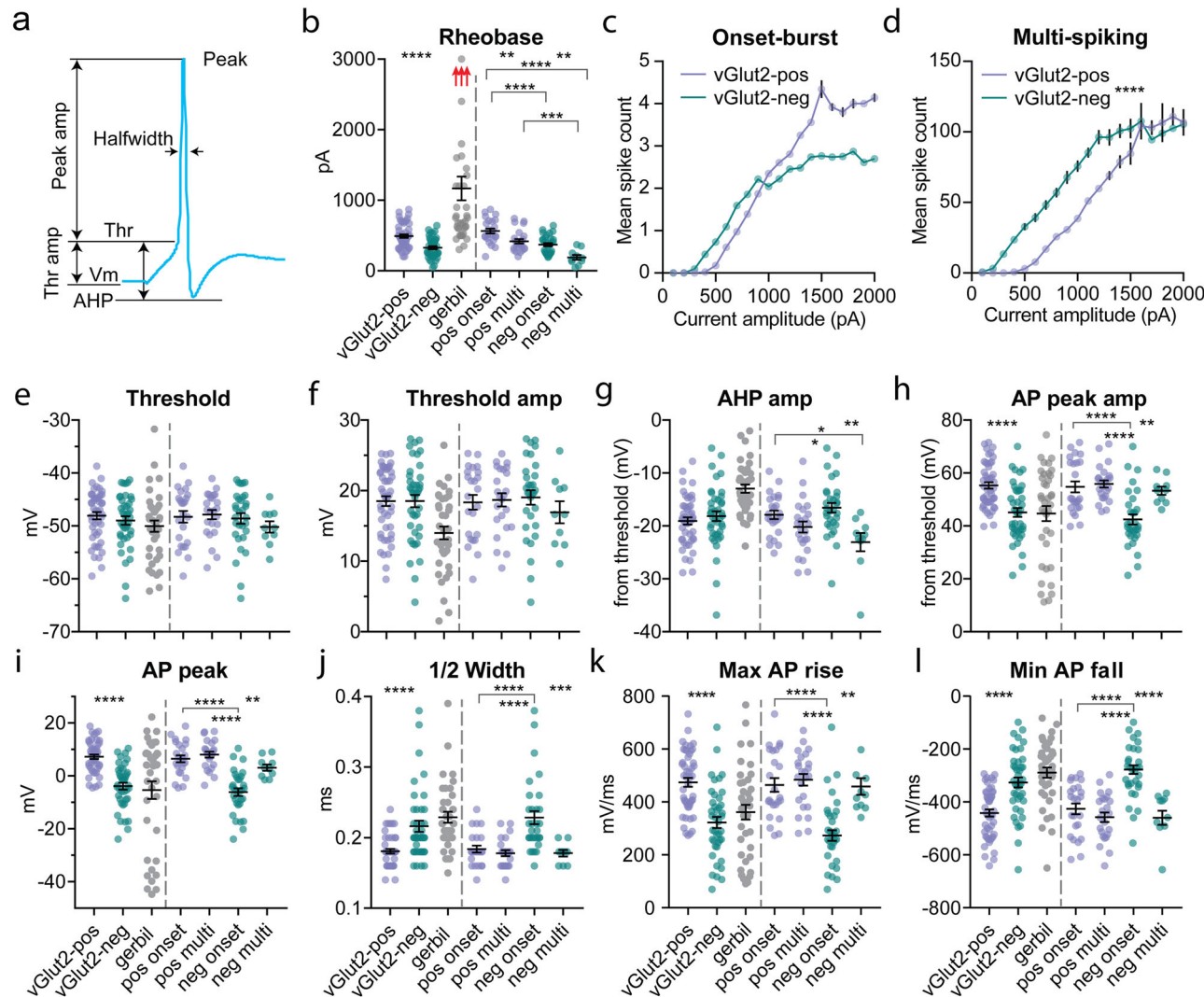

**Fig. 2 Inhibitory LSO PNs have lower activation thresholds and excitatory LSO PNs have larger, faster action potentials. a** Diagram illustrating AP parameter measurements. **b** Rheobase. Red arrows indicate out-of-range gerbil rheobases up to 4.2 nA. **c**, **d** Mean spike counts in 100 pA current bins for excitatory and inhibitory LSO neurons separated by firing type. Error bars smaller than symbols are omitted. $n$ = cells(animals): vGlut2-pos onset: 19(17), vGlut2-neg onset: 30(21), vGlut2-pos multi: 24(19), vGlut2-neg multi: 10(9). **e** AP threshold. **f** Threshold amplitude. **g** Afterhyperpolarization amplitude. **h** AP peak amplitude. **i** AP peak voltage. **j** AP half-width. **k** AP maximum rise slope. **l** AP minimum fall slope. For (**b** and **e–l**) $n$ = cells(animals) from left to right: 48(30), 41(24), 38(22), 24(19), 24(19), 31(21), 10(9). Mean ± SEM. For pooled transmitter type data (left side bars) unpaired two-tailed $t$-test is shown. For firing type separated groups (right side bars), Tukey corrected multiple comparisons of two-way ANOVA is shown. *$p < 0.05$, **$p < 0.01$, ***$p < 0.001$, ****$p < 0.0001$.

significant interaction of firing and transmitter type ($F(1,85) = 5.588$, $p = 0.0204$). The interaction accounted for 4.18% of the total variance while transmitter and firing type accounted for 12.67% and 7.13% respectively. For minimum AP fall, there were main effects of transmitter ($F(1,85) = 10.72$, $p = 0.0015$, Fig. 2l right) and firing type ($F(1, 85) = 18.97$, $p < 0.0001$), however, there was a significant interaction of firing and transmitter type ($F(1, 85) = 8.705$, $p = 0.0041$). The interaction accounted for 6.24% of the total variance while transmitter and firing type accounted for 7.68% and 13.59% respectively. Post hoc analysis suggests that inhibitory onset-burst LSO PNs drove differences with slower AP kinetics.

We also asked if there was any influence of tonotopic location or sex on intrinsic membrane properties and AP parameters in mice. We divided the LSO by bisecting the outline of the LSO from tip to tip to create a curving midline then dividing the nucleus into 3 equal parts perpendicular to midline to crudely represent high (medial), middle, and low-frequency (lateral)

regions[25]. We then analyzed all parameters using four-way ANOVA and found no interaction between tonotopy and sex, so we performed separate three-way ANOVAs to reduce complexity. We found that there were no main effects of tonotopy nor interactions between firing type and tonotopy, however, there was an interaction between transmitter and tonotopy for input resistance only (Fig. S2 and Tables S1 and 2). Post hoc analysis of the input resistance groups found differences across transmitter types and tonotopic locations, however, this interaction may have been influenced by the presence of an inhibitory subgroup with only one cell (low-frequency inhibitory multi-spiking, Fig. S2). These data indicate that there was no substantial tonotopic gradients or associations for intrinsic membrane or AP properties.

With regard to sex, we only found effects/interactions for sag potential (Fig. S3 and Tables S3 and 4). For sag, there was a main effect of sex and an interaction between firing type and sex (Fig. S3 and Tables S3 and 4), however, post hoc analysis revealed

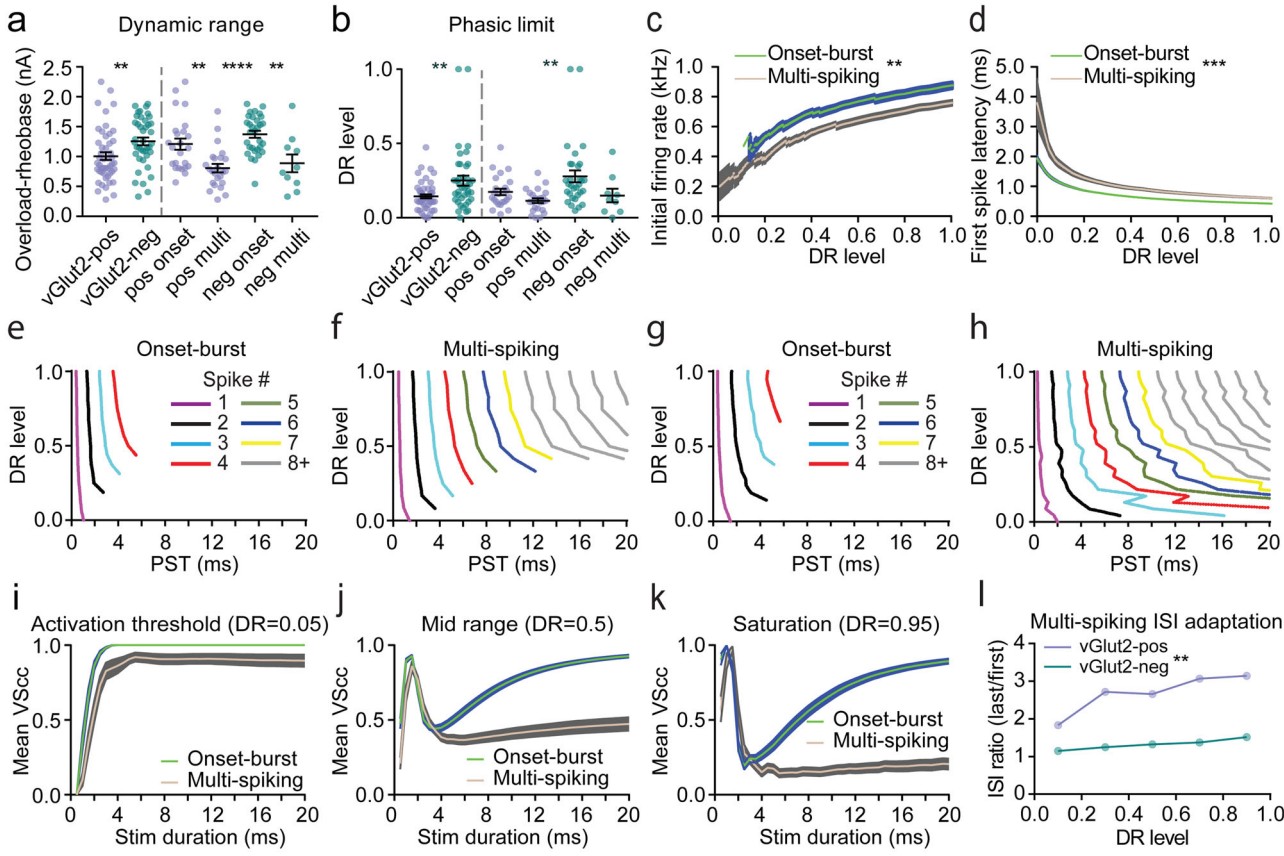

**Fig. 3 Onset-burst and multi-spiking neurons similarly encode timing at the onset of stimuli, but onset-burst cells maintain this ability across a wider range of stimulus intensities and durations. a** Dynamic range from rheobase to overload current. $n$ = cells(animals) from left to right: 48(30), 41(24), 24(19), 24(19), 31(21), 10(9). **b** Phasic limit on DR scale. $n$ = cells(animals) from left to right: 48(30), 41(24), 24(19), 24(19), 31(21), 9(9). **c** Initial firing rate by DR level ±SEM for onset-burst and multi-spiking cell types. $n$ = cells(animals): onset 49(25), multi 34(24). **d** First spike latency by DR level ±SEM. $n$ = cells(animals): onset 49(25), multi 34/(24). Example excitatory onset-burst (**e**) and multi-spiking (**f**) neurons DR level by interpolated post stimulus onset time (PST) of APs color coded by sequential order. **g, h** Example inhibitory neurons as in (**e**) and (**f**). **i** Average of individual neuron cycle-by-cycle vector strength (VScc) for both firing types at DR level 0.05 (rheobase). **j** As in (**i**) but at DR level 0.5 (mid-range). **k** As in (**i**) but at DR level 0.95 (saturation). **l** Interspike interval ratio (avg last 3 to avg first 3) binned at 0.2-DR level for inhibitory and excitatory multi-spiking neurons. Error bars smaller than symbols are omitted. $n$ = cells(animals): vGlut2-pos multi: 24(19), vGlut2-neg multi: 10(9). Mean ± SEM. For pooled transmitter type data (left side bars) unpaired two-tailed $t$-test is shown. For firing type separated groups (right side bars), Tukey corrected multiple comparisons of two-way ANOVA is shown. *$p < 0.05$, **$p < 0.01$, ***$p < 0.001$, ****$p < 0.0001$.

no differences between groups based on firing type and sex. Additionally, the interaction may be influenced by there being no inhibitory multi-spiking cells in the female group. These data indicate that sex did not substantially influence intrinsic properties of LSO PNs.

We also tested the grouping of LSO PNs using a less biased exploratory approach. We used principal component analysis (PCA) of intrinsic membrane and AP properties followed by k-means clustering (Fig. S4). This analysis suggested there were two groups and that transmitter type explained most of the clustering (Yule's correlation between transmitter type and cluster, $\varphi = -0.85$, $p < 0.0001$, permutation test for correlation, Yule's correlation between firing type and cluster, $\varphi = 0.18$, $p = 0.14$, permutation test for correlation, Fig. S4). Similar to the analyses presented above, these data indicate that transmitter type is the dominant factor differentiating LSO PNs intrinsic membrane properties.

**Time coding and integrative ability differences between LSO PN firing types.** Inhibitory LSO PNs in mice had a wider dynamic range measured from rheobase to overload current

where spiking decreased ($p = 0.0088$, $t$-test, Fig. 3a left). Overloading was accompanied by abruptly decreasing spike amplitudes leading to spike failures toward the end of a sweep presumably due to inability of voltage-gated sodium channels to recover from inactivation. When we analyzed dynamic range by firing type as well using two-way ANOVA, there was no main effect of transmitter ($F_{(1,85)} = 1.883$, $p = 0.1736$, Fig. 3a right) however, there was a main effect of firing type ($F_{(1,85)} = 24.80$, $p = 0.0001$) and no interaction ($F_{(1,85)} = 0.2099$, $p = 0.6480$). These data indicate that dynamic range is largely determined by firing type and suggest that difference between pooled transmitter types may be due to a preponderance of onset-burst cells in the inhibitory group.

We also analyzed properties of LSO PNs that may relate to time and level coding ability. Firing type has long been thought to impact feature extraction and thus was the focus of our comparisons, however, both firing types were found in both transmitter types and there were large differences between transmitter types in intrinsic membrane excitability. Therefore, we normalized the responses of mouse LSO PNs into a dynamic range (DR) scale from rheobase at DR = 0 to current overload at DR = 1. The underlying rationale behind this approach was that

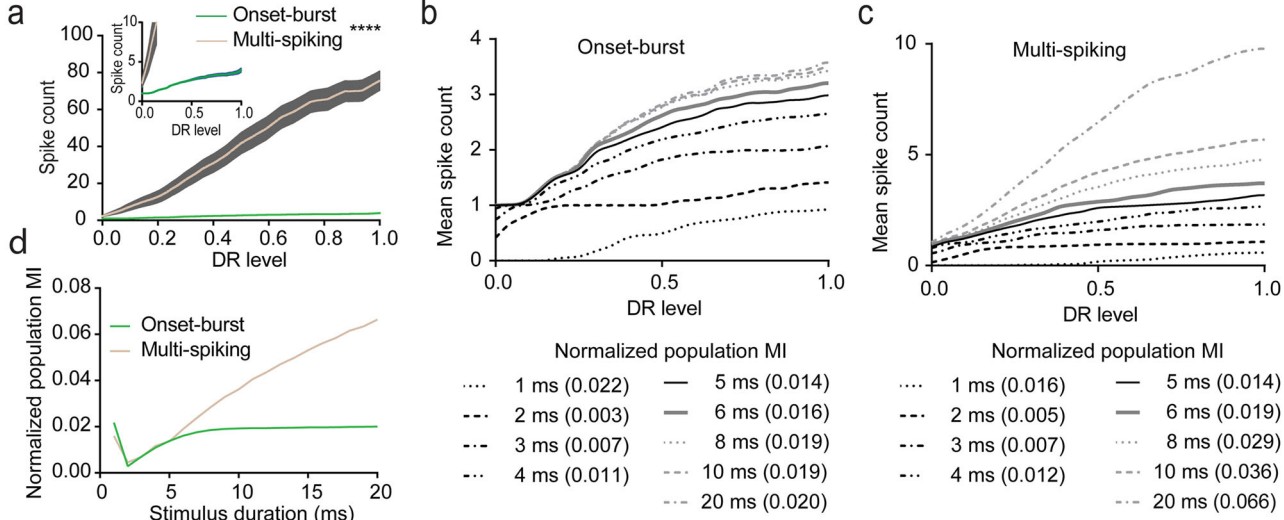

**Fig. 4 Onset-burst neurons may encode some level information for brief stimuli and multi-spiking neurons excel at level coding over a wider range of stimulus intensities and durations. a** Average spike counts by DR level ±SEM for onset-burst and multi-spiking LSO PNs. Inset shows expanded scale near origin. n = cells(animals): onset 49(25), multi 34(24). **b** Mean spike counts vs. DR levels for onset-burst population. Each curve represents population mean spike counts computed for a stimulus duration ranging from 1 through 20 ms. Legends show stimulus duration in ms and normalized mutual information (NMI) between population-mean spike counts and DR level. n = cells(animals): 49(25). **c** Same as in (**b**), but for multi-spiking population. n = cells(animals): 34(24). **d** NMIs for different stimulus durations between firing types. n = cells(animals): onset 49(25), multi 34(24). Stimulus durations interpolated at 1 ms. ****$p < 0.0001$.

the cumulative strength of a neuron's synaptic inputs is scaled to its intrinsic excitability. With this assumption, DR level may be a useful stand-in for relative sound level in LSO PNs.

Near rheobase, virtually all LSO PNs exhibited truly phasic onset responses with a single spike that could encode timing maximally. We found that inhibitory LSO PNs maintained their phasic firing to higher DR levels ($p = 0.0028$, $t$-test, Fig. 3b left). When we analyzed phasic limit by firing type as well using two-way ANOVA, again there was no main effect of transmitter ($F(1,84) = 3.550$, $p = 0.0630$, Fig. 3b right) however, there was a main effect of firing type ($F(1,84) = 6.441$, $p = 0.0130$) and no interaction ($F(1,84) = 0.9089$, $p = 0.3431$). These data suggest firing type is the primary driver of phasic limit.

On the DR scale, initial firing rate (IFR) was higher (main effect of firing type, $\beta = -0.1$, standard error in coefficient estimate (SE) $= 0.04$, $p = 0.0069$, linear mixed model (LMM) $t$-test for coefficient where $t$-statistic $= \beta/SE$, interaction of firing type and transmitter type, $\beta = -0.14$, SE $= 0.05$, $p = 0.0092$, LMM $t$-test for coefficient, Fig. 3c) and first spike latency (FSL) was shorter in the onset-burst cells (main effect of firing type, $\beta = 0.94$, SE $= 0.22$, $p < 0.0001$, LMM $t$-test for coefficient, interaction of DR level and firing type, $\beta = -1.07$, SE $= 0.29$, $p = 0.0002$, LMM $t$-test for coefficient, Fig. 3d). We asked if the FSL spread at DR level of 0 was explained by intrinsic differences. FSL had a modest negative correlation with rheobase magnitude (Spearman's $\rho = -0.37$, $p = 0.0004$, $t$-test for correlation), however, there was a robust correlation of FSL with membrane time constant (Spearman's $\rho = 0.67$, $p < 0.0001$, $t$-test for correlation) suggesting that differences in membrane time constant rather than rheobase largely underlie FSL variability. Jitter in FSL could also affect time coding ability. Since no current step was given more than once, this parameter could not be estimated for individual neurons. However, consistency of FSL in the two populations can be compared by using the population standard deviation of this parameter. At rheobase/DR = 0, the standard deviation of onset-burst population was lower (Onset: 0.9 ms, Multi: 4.99 ms, $p < 0.0001$, 2-sample $F$-test), suggesting jitter may be lower in onset-burst cells.

Differences in firing types that could affect time encoding can be seen in the interpolated spike times of the first spike, second spike, and so on (Fig. 3e–h). At higher intensities, a deviation from phasic response with systematic addition of larger number of spikes may further degrade timing information contained in individual spike times in a multi-spiking cell (Fig. 3f, h) compared to an onset-burst cell (Fig. 3e, g). To test this further, we calculated the cycle-by-cycle vector strength (VScc) for each neuron within the truncated stimulus durations from 0 to 20 ms at DR levels 0.05, 0.5, and 0.95. Near rheobase/DR = 0 in both firing types, average VScc increased rapidly with increasing pulse width as single spikes afford consistent timing information (Fig. 3i) suggesting that both firing types have high time encoding ability near rheobase. As DR level increases, VScc erodes in both firing types with longer stimulus durations as additional spikes introduce timing variability through spike-phase dispersion, however, VScc recovers in onset-burst cells as the number of spikes stabilizes near the onset and remains constant as pulse width increases (Fig. 3j, k). These data suggest that onset-burst LSO PNs may be able to encode timing over a wider range of stimulus durations.

Inter-spike interval (ISI) was more consistent across the 200 ms duration of current injections and across DR levels in inhibitory LSO PNs (main effect of neurotransmitter type, $\beta = -5.85$, SE $= 1.995$, $p = 0.004$, LMM $t$-test for coefficient, interaction of DR level and neurotransmitter type, $\beta = 4.71$, SE $= 1.74$, $p = 0.007$, LMM $t$-test for coefficient, Fig. 3l). The ramifications of spike adaptation on integration of ongoing level information are not known, however, if firing rate is expected to encode location, then rate changes during static stimuli may degrade the signal.

To probe the integrative abilities of LSO PNs that may be beneficial for encoding sound intensity, we analyzed interpolated spike counts over DR level. Multi-spiking cells had steeper increases in spike number, and thus sensitivity, even for initial spiking at low DR levels (main effect of firing type, $\beta = -4.72$, SE $= 2.75$, $p = 0.086$, LMM $t$-test for coefficient, interaction of DR level and firing type, $\beta = 77.34$, SE $= 5.11$, $p < 0.0001$, LMM $t$-

test for coefficient, Fig. 4a inset). Also, multi-spiking cells continue to accumulate spikes at higher DR levels whereas onset-burst cells quickly plateau (Fig. 4a).

As a measure of level coding strength, we calculated normalized population mutual information (NMI) between population-mean spike counts and DR level. Onset-burst and multi-spiking LSO PNs had similar DR level coding in their population mean spike count for the initial part of the current step (Fig. 4b, c). At the shortest tested stimulus duration of 1 ms, a systematic decrease in first spike latency with DR level increased the probability of inclusion of a spike into the 1 ms window, causing a higher initial NMI (Fig. 4d). Less systematic variation in the inclusion of a second spike with increasing DR level within the temporal windows of 2 and 3 ms lowered the NMI in both populations. For longer stimulus durations, NMI increased for both onset-burst and multi-spiking cell responses due to inclusion of more spikes with increasing stimulus duration. NMI plateaued for onset-burst cells as responses were confined to the first 8 ms (Fig. 4d) whereas multi-spiking cells continued to accrue information for the entire 200 ms duration reaching a max NMI = 0.19. Together, analysis by DR level data provides confirmatory details for superior level coding by multi-spiking LSO PNs.

**Dendritic morphology differs between LSO PN transmitter types.** The majority of patch-clamped LSO PNs in mice ($n = 70$) were imaged using two-photon microscopy immediately after electrophysiological recordings. This allowed us to reconstruct the dendritic arbor and roughly place cells within the LSO without fixation/shrinkage artifacts. Most cells exhibited fusiform/bipolar shape often with one dendrite wrapping back toward the long axis of the other (Fig. 5a–d) similar to prior reports of LSO PN morphology[33–37]. Most LSO PNs had centered soma with 68% of cells having an eccentricity below 0.5 (Fig. 5g, see "Methods"). All cell types displayed a range of eccentricity values; however, most cells fell at either end of the spectrum suggesting that their dendrites usually spread evenly in both directions (eccentricity near 0) or all go in one direction (eccentricity near 1) likely along the frequency lamina. Eccentricity was not related to the distance to LSO boundary ($R^2 = 0.0012$, Fig. 5h).

Patch-clamp recordings were made from identified cells clearly within the border of the LSO that appeared healthy and met PN criteria (see methods), but without intent to thoroughly sample the tonotopic range. In mice, all cell types were more commonly recorded in the mid-LSO ($\chi^2$, equal proportions, $[2,n = 3] = 11.96$, $p = 0.0025$, Fig. 5e i). Uneven distributions of onset-burst and inhibitory cell groups (onset-burst: $\chi^2$, equal proportions, $[2,n = 3] = 13.5$, $p = 0.0012$; inhibitory: $\chi^2$, equal proportions, $[2,n = 3] = 11.76$, $p = 0.0028$; Fig. 5e i) were due to a mid-LSO bias of inhibitory onset-burst cell-type (61% mid-LSO, $\chi^2$, equal proportions, $[2,n = 3] = 11.68$, $p = 0.0029$, Fig. 5e ii).

In gerbils, available morphological/distribution information was limited to observations made at the time of recordings regarding shape and location within the LSO. True phasic firing LSO PNs were most often encountered in the lateral/low-frequency limb (6/12, 50%, $\chi^2$, equal proportions, $[2,n = 3] = 2$, $p = 0.368$), while onset-burst and multi-spiking LSO PNs were found predominantly in medial locations (O: 12/18, 67%, $\chi^2$, equal proportions, $[2,n = 3] = 9$, $p = 0.011$: M: 5/9, 56%, $\chi^2$, equal proportions, $[2,n = 3] = 4.75$, $p = 0.093$). Multi-spiking LSO PNs were found throughout the body of the LSO in gerbils and did not appear to have overtly distinctive dendritic features from other firing types.

In mice, inhibitory and excitatory LSO PNs were found at similar distances from the nearest boundary of the LSO ($p = 0.15$,

$t$-test, Fig. 5f left). In two-way ANOVA, there were no main effects of transmitter ($F_{(1,66)} = 0.7125$, $p = 0.4017$, Fig. 5f right) or firing type ($F_{(1,66)} = 2.155$, $p = 0.1468$) or interaction ($F_{(1,66)} = 2.199$, $p = 0.1429$). Multi-spiking cells were observed throughout the body of the LSO. Approximately 14% of cells (10/70) had at least one dendritic tip that crossed the LSO boundary. However, these cells' somata were not significantly closer to the boundary than other cells ($p = 0.102$, $t$-test).

The tips of reconstructed dendritic arbors in mice exhibited more spread in the coronal plane than along the rostro-caudal/$z$-axis and there were no differences between transmitter ($xy/z$ spread, I: $6.54 \pm 0.53$, E: $7.65 \pm 0.6$, $p = 0.19$, $t$-test) or firing types ($xy/z$ spread, O: $7.33 \pm 0.61$, M: $7.4 \pm 0.61$, $p = 0.71$, $t$-test). However, the $z$ axis dimensions may be reduced by loss of dendrites at the surface of the brain slice and lower imaging resolution with depth. Dendrite loss with slicing was not analyzed, however, since cells appeared healthy at the time of recording, major dendrite loss would not be expected as such damage often leads to cell death during recovery from slicing.

Although our methods did not allow us to visualize the complex internal tonotopic axis of the mouse LSO, we made an effort to explore cell-type differences in dendritic spread across isofrequency bands. We chose cells for which dendritic extension from soma was at least 55 µm and an eccentricity of at most 0.9. This yielded 50 neurons of which 44 had a dendritic orientation axis that visually aligned within a presumed isofrequency band. The six remaining neurons had an orientation axis such that it would visually appear to cross multiple isofrequency bands. These neurons were of either firing type (4 onset-burst, 2 multi-spiking) and neurotransmitter types (2 excitatory, 4 inhibitory). For these frequency-crossing neurons, an approximate estimate of presumptive cross-frequency spread was obtained as the maximum dendritic spread along mediolateral axis that deviates from the actual tonotopic axis[33]. Frequency spread ranged from 19.7 µm to 223.7 µm (avg $82.5 \pm 5.6$ µm). This spread was not significantly different between either firing (onset-burst: $81.9 \pm 7.6$ µm, multi-spiking: $98.7 \pm 10.8$ µm, $p = 0.20$, $t$-test) or neurotransmitter types (inhibitory: $79.3 \pm 8.5$ µm, excitatory: $94.8 \pm 8.8$ µm, $p = 0.24$, $t$-test) and did not correlate with distance from the nearest boundary (Spearman's $\rho = -0.051$, $p = 0.723$, $t$-test for correlation) or eccentricity (Spearman's $\rho = 0.05$, $p = 0.734$, $t$-test for correlation). These observations held even when excluding the 6 cross-tonotopically oriented neurons from analysis. These results suggest that frequency spread of LSO PNs is not predicted by cell-type or dendritic shape.

We found that inhibitory and excitatory LSO PNs had similar soma volume ($p = 0.594$, $t$-test, Fig. 5i left), primary dendrite diameter ($p = 0.631$, $t$-test, Fig. 5j left), and maximum dendritic extension ($p = 0.22$, $t$-test, Fig. 5k left). Soma volume was not correlated with input resistance (Spearman's $\rho = -0.16$, $p = 0.19$, $t$-test for correlation) or rheobase (Spearman's $\rho = 0.03$, $p = 0.79$, $t$-test for correlation). Transmitter types did differ in their total dendritic length ($p = 0.007$, $t$-test, Fig. 5l left), number of dendritic branch points ($p = 0.025$, $t$-test, Fig. 5n left), and number of primary dendrites ($p = 0.013$, $t$-test, Fig. 5m left) all of which were larger in excitatory LSO PNs. Excitatory LSO PNs also exhibited wider ranges in these categories and the most expansive cells were excitatory.

We also examined the influence of transmitter and firing type together on morphological parameters using two-way ANOVA. For soma volume, there were no main effects of transmitter ($F_{(1,66)} = 0.09169$, $p = 0.7630$, Fig. 5i right) or firing type ($F_{(1,66)} = 2.109$, $p = 0.1511$) or interaction ($F_{(1,66)} = 0.5476$, $p = 0.8157$). For primary dendrite diameter, there were no main effects of transmitter ($F_{(1,66)} = 0.6064$, $p = 0.4389$, Fig. 5j right) or firing type ($F_{(1,66)} = 1.476$, $p = 0.2287$) or interaction

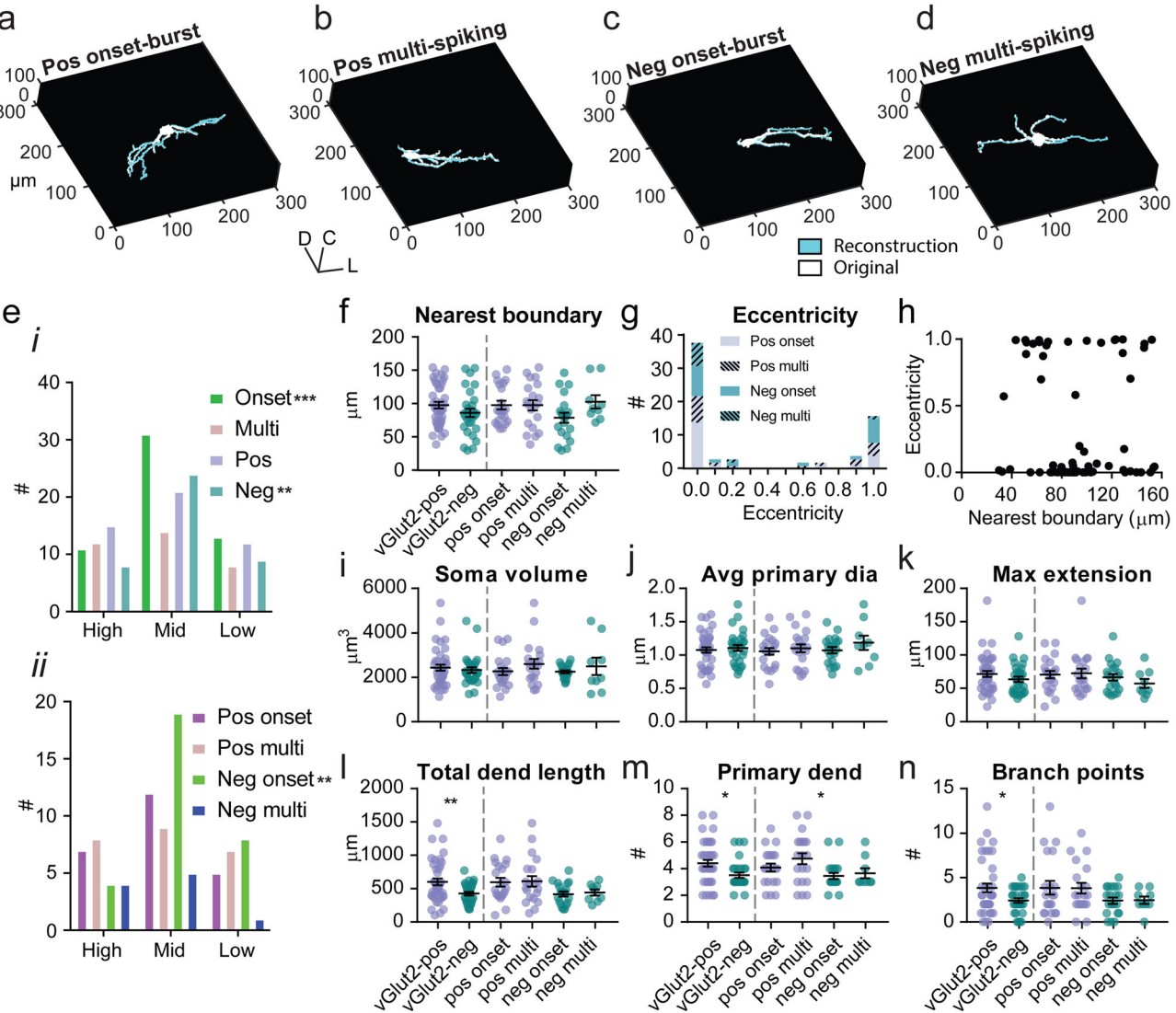

**Fig. 5 Excitatory LSO PNs have more complex dendritic arbors.** Example 3D renderings of reconstructions for vGlut2 positive (excitatory, **a**, **b**) and vGlut2 negative (inhibitory, **c**, **d**) LSO PNs. **e** i, ii Neuron counts for cell-type categories divided into crude high (medial), middle, and low (lateral) frequency regions of the LSO. **f** Distance from soma to nearest LSO boundary. **g** Neuron counts for cell-type categories for eccentricity computed using the cosine of the maximum angle between tip-to-soma-center vectors with centered soma configuration set to 0 and eccentric soma configuration having a value of 1. **h** Eccentricity plotted relative to distance from soma to nearest boundary. **i** Soma volume. **j** Average primary dendrite diameter. **k** Maximum extension: length of a straight-line connecting origin of a dendrite and its farthest tip. **l** Total dendritic length: sum of lengths of all primary dendrites and dendritic branches. **m** The number of primary dendrites. **n** Total number of branch points. For (**f**) and (**i–n**) $n$ = cells(animals) from left to right: 41(25), 29(18), 21(16), 20(15), 20(16), 9(9). Mean ± SEM. For pooled transmitter type data (left side bars) unpaired two-tailed $t$-test is shown. For firing type separated groups (right side bars), Tukey corrected multiple comparisons of two-way ANOVA is shown. *$p < 0.05$, **$p < 0.01$, ***$p < 0.001$, ****$p < 0.0001$.

($F_{(1,66)} = 0.2659$, $p = 0.6078$). For maximum dendritic extension, there were no main effects of transmitter ($F_{(1,66)} = 2.077$, $p = 1543$, Fig. 5k right) or firing type ($F_{(1,66)} = 0.3302$, $p = 0.5675$) or interaction ($F_{(1,66)} = 0.6878$, $p = 0.4099$).

For total dendritic length, there was a main effect of transmitter ($F_{(1,66)} = 6.558$, $p = 0.0127$, Fig. 5l right) but not firing type ($F_{(1,66)} = 0.07206$, $p = 0.7892$) nor interaction ($F_{(1,66)} = 0.006531$, $p = 0.9358$). For number of primary dendrites, there was also a main effect of transmitter ($F_{(1,66)} = 5.584$, $p = 0.0211$, Fig. 5m right) but not firing type ($F_{(1,66)} = 1.419$, $p = 0.2378$) nor interaction ($F_{(1,66)} = 0.3587$, $p = 0.5513$). For number of dendritic branch points, there was again a main effect of transmitter ($F_{(1,66)} = 4.578$, $p = 0.0361$, Fig. 5n right) but not firing type ($F_{(1,66)} < 0.0001$, $p = 0.9923$) nor interaction ($F_{(1,66)} = 0.0.005972$, $p = 0.9386$). Together,

these data indicate that transmitter type is the main driver of differences in dendritic complexity.

We also tested whether there was any influence of tonotopic location or sex on morphological properties of LSO PNs. We analyzed all parameters using four-way ANOVA and found no interaction between tonotopy and sex, so we performed separate three-way ANOVAs to reduce complexity. For tonotopy, we found that there were no main effects or interactions (Fig. S5 and Tables S5 and 6) indicating that there was no substantial tonotopic gradients or associations for morphological properties.

With regard to sex, we found a main effect of transmitter on total dendritic length, a main effect of firing type on maximum dendritic extension, and interaction of firing type and sex on soma volume, max dendritic extension, total dendritic length and primary dendrite diameter (Fig. S6 and Tables S7 and 8).

However, post hoc analysis did not reveal any differences between groups based on firing type and sex. Additionally, the lack of inhibitory multi-spiking cells in the female group may have influenced interaction between firing type and sex. Nonetheless, these data indicate that sex did not overtly influence morphological properties of LSO PNs.

We also used principal component analysis of morphological properties followed by k-means clustering (Fig. S7). This analysis again suggested there were two groups, and that transmitter type explained the clustering better than firing type (Yule's correlation between transmitter type and cluster, $\varphi = 0.34$, $p = 0.008$, permutation test for correlation, Yule's correlation between firing type and cluster, $\varphi = -0.05$, $p = 0.91$, permutation test for correlation, Fig. S7).

## Discussion

We generally found more and larger differences in intrinsic membrane properties between transmitter types than firing types, potentially allowing differentially extracted information to be segregated by transmitter system in higher processing centers such as the IC. Inhibitory LSO PNs exhibited higher input resistances (61% difference) with correspondingly lower rheobase as well as higher spike output (Fig. 2b–d). However, they were also on average ~5 mV more hyperpolarized at rest and displayed a wider range of hyperpolarized resting membrane potentials (Fig. 1e). Greater hyperpolarization did not outweigh differences in input resistance since at resting membrane potential the input resistance difference was even larger and inhibitory LSO PNs continued to have substantially lower rheobase. These data suggest that inhibitory LSO PNs have lower activation threshold. This configuration may provide a source of early inhibition that has been observed in IC neurons in vivo[38–40]. Additionally, the inhibitory pathway from LSO to IC is largely or entirely segregated to the ipsilateral side[12,16,25,41–45], thus, preceding either in time or relative level domain, ipsilateral inhibition may promote consolidation of contralateral auditory object representation. Interpretation of the intrinsic excitability differences between inhibitory and excitatory LSO PNs could be impacted by the relative magnitude of cell-type-specific synaptic inputs which will be the focus of future studies. It has also been shown that ipsilateral and contralateral projections from LSO to IC target different bands or territories in cats[46] suggesting inhibitory and excitatory drive may participate in sub-circuits within the IC.

Square pulse current injections have long been used to probe the response properties of neurons ex vivo[11] and it was recently shown using in vivo whole-cell recordings that such current injections closely mimic sound evoked responses in LSO neurons[7]. We found that all LSO PNs exhibited phasic firing near activation threshold but in mice separate into 2 firing types at higher current injection levels, onset-burst and multi-spiking (Figs. 1 and 6c). Using our DR level analysis (Fig. 3), we demonstrated that onset-burst LSO PNs may excel at encoding timing, and they had larger dynamic ranges (Fig. 3a) potentially allowing them to encode timing over a larger range of stimulus levels without regard to stimulus duration. Conversely, DR level analysis of level information (Fig. 4) showed that multi-spiking LSO PNs could provide individually integrable level information across a wider response range.

There is disagreement in the literature regarding LSO PNs firing types. To encode relative intensity, LSO PNs were expected to fire in a graded manner in which spiking rate increases with intensity[6]. Indeed, in vivo reports corroborated this assumption, finding primarily "chopper" multi-spiking responses[4–6,9,47–49], although some onset/phasic responses were observed[9,49]. Ex vivo studies have reported multi-spiking only[35,50–52], a combination of

onset and multi-spiking[24,53,54] that in some cases exhibited tonotopic gradients[36,55] or even onset responses only[56,57]. The reasons for differences in LSO PNs firing type findings are not clear, however, expectations and experimental conditions such as species, age, solution composition, and temperature may play a role. We found that some (9) LSO PNs changed firing type after breaking into whole-cell mode. There were slightly more multi-to-onset (6) than onset-to-multi (3) shifts, however, this would not seem to be a major driver of differences between studies. Nonetheless, these data suggest that firing type may be modulable. Phasic, onset-burst, and multi-spiking firing types are similar to the spectrum of firing responses that can be produced by varying voltage-gated potassium channel (VGPC) conductance in modeling studies of ventral cochlear nucleus neurons[58] and VGPCs can be modulated by a variety of factors[59] including osmolarity[60] and temperature[61]. Species differences in the relative number of firing types could also reflect differential utilization of ILD/ITD information for ecological or biomechanical reasons.

The most recent in vivo reports utilized whole-cell technique in gerbils and also found onset and multi-spiking LSO neurons[7,8]. These authors found that multi-spiking cells were near the margins of the nucleus, and they tended to have lower somatic synapse coverage suggesting they were not PNs but the marginal cells described by Helfert et al.[62]. The onset cells tended to have greater somatic synapse coverage and were proposed to be the PNs designated by Helfert which made up 79% of LSO neurons. Franken et al. also found that the LSO neurons with onset firing responses had short spike amplitudes normally associated with time-coding-specialized medial superior olive (MSO) neurons[63] and excelled at encoding of transients[8]. Thus, the authors proposed that a large majority of LSO neurons are time-coding-specialized, and the dominant function of LSO lies with ITD coding.

It is not known whether degree of somatic synapse coverage has functional significance for sound localization or association with projection pattern in the LSO. Prior studies of LSO PN intrinsic membrane properties, and our current study, did not examine synapse coverage[24,50,51,53–57]. A few prior studies examined dendritic morphology along with firing responses[35,36], however, morphology was not used as a factor in determining which cells were considered LSO PNs.

Ex vivo in gerbils, we observed phasic, multi-spiking and onset-burst firing types that did not appear overtly different in distance to LSO boundary or dendritic morphology. Some of the phasic cells were indeed MSO-like[64,65] with very low input resistances and small spikes (Figs. 1f and 2i), however, phasic cells were only 31% of those recorded. These cells were not found in mice and were biased to the low-frequency limb in gerbils suggesting an association with low-frequency hearing but making it unlikely that they are required for overall LSO function. Additionally, high-frequency adapted animals similar to mice may have been the ancestral state[66] making it unlikely that such properties were an obligatory part of the evolution of this circuit. Muti-spiking and onset-burst LSO PNs in mice did not differ in distribution, location, or morphology suggesting that multi-spiking LSO PNs are not marginal cells.

While our firing-type results would seem to paint a more balanced, multi-functional picture, we also found that near activation threshold all LSO PNs neurons exhibited phasic/onset firing of single APs (Figs. 1d and 6c). This suggests that at this sensitive transition point, the entire population would be able to optimally encode timing of onsets, transients, or amplitude modulations, but not provide individually integrable level information regardless of stimulus duration. Thus, there may be 2 functional modes for the LSO. When a sound source is near the transition from inhibition to excitation, LSO PNs are mainly

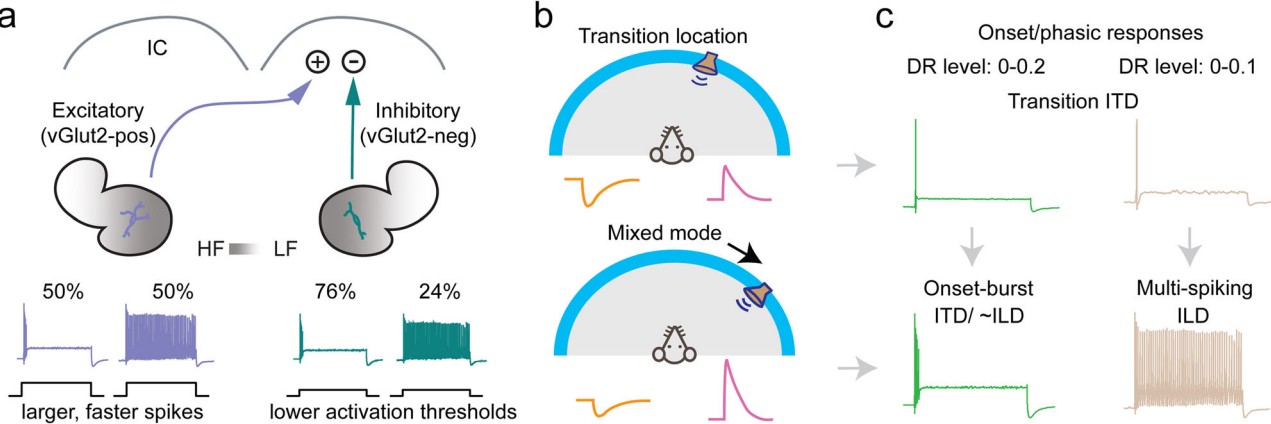

**Fig. 6 Illustrated major conclusions. a** Inhibitory LSO PNs project to IC ipsilaterally whereas excitatory cells project contralaterally. LSO PN types exhibited similar tonotopic distribution and had both onset-burst and multi-spiking firing types. Excitatory cells had more complex dendritic arbors and larger, faster APs. Inhibitory cells had lower activation threshold. **b** Top: illustration of a sound source location and synaptic drive where an LSO PN would be transitioning from being inhibited to firing. At this point all cells fire phasically (single spikes). Bottom: for the same neuron, a source location further toward the ipsilateral ear results in greater dominance of excitation and mixed firing modes within the population. **c** Top: illustration of 2 cells similarly tuned for a particular location with the sound source near their transition location (**b** top) where both cell types exhibit phasic firing responses that may favor time encoding functions or provide single spike level transition information. The phasic limit on dynamic range (DR) scale was higher for onset-burst cells. Bottom: as the sound source moves toward the ipsilateral ear (**b** bottom) there is greater excitation and onset-burst and multi-spiking responses may emerge. Multi-spiking cells allow for more effective integrative level difference functions while onset-burst cells retain the ability to robustly encode timing regardless of stimulus duration.

providing timing information (Fig. 6b, c top). Since for any given azimuth position there may only be a small subset of cells at this sensitive transition point, all LSO PNs having this ability would make this mode more robust.

As a sound source moves toward the ipsilateral ear, excitation becomes increasingly dominant and the onset-burst and multi-spiking firing types may emerge allowing for more effective integration of relative intensity (Fig. 6b, c bottom). Onset-burst cells fire a small number of spikes and although this could encode some intensity information, even for brief stimuli (Fig. 4d), it is unknown whether this can be utilized in higher processing centers. Onset-burst LSO PNs do however continue to be able to encode timing robustly even as stimulus duration and intensity increases (Fig. 3j, k). Thus, together our analysis of spiking by DR level demonstrates how diversity in the LSO PN population may expand the range of sound intensities and stimulus durations over which the LSO can encode timing and level information.

Although both transmitter systems included both firing types, inhibitory LSO PNs were more likely to be onset-burst (76%) and have higher phasic limit (Fig. 3b), lower maximum number of spikes (Fig. 2c), and more stable ISI (Fig. 3l)—all of which could provide advantages for time coding. This may imply that ipsilateral inhibition to the IC may favor time coding functions allowing for the segregation of this information. Potential support for such segregation comes from a recent study showing that ITD processing for amplitude modulations (more likely encoded by LSO) and fine structure (more likely encoded by MSO) take place in different IC regions[67].

Excitatory LSO PNs on the other hand, had a comparatively larger number of multi-spiking cells (50%) and had onset-burst cells with higher maximum number of spikes (Fig. 2c) both of which could confer advantages for integrative intensity coding. They also had more complicated dendritic arbors (Fig. 5l–n). Since inhibitory inputs to LSO PNs target their soma and proximal dendrites[68], more complicated dendritic arbors may house more excitatory inputs. If this is the case, excitatory LSO PNs in general may be better tuned for integrative intensity functions at higher activation levels.

Although questions remain, together these data bring a clearer picture of cellular diversity in the LSO that may allow for the extraction of different types of sound information and segregation of that information in higher processing centers.

## Methods

**Animals**. All animal procedures were approved by the Northeast Ohio Medical University Animal Care and Use Committee in accordance with the guidelines of the United States National Institutes of Health. Mice were procured from Jackson Labs (Bar Harbor, ME) and Mongolian gerbils were procured from Charles River Laboratories (Wilmington, MA). We produced vGlut2 reporter mice by crossing a vGlut2-cre mouse line (B6J.129S6[FVB]-Slc17a6[tm2(cre)Lowl]/MwarJ; RRID: IMSR_JAX:028863) with Ai9 tdTomato reporter mice (B6.Cg-Gt(ROSA)26Sort[m9(CAG-tdTomato)Hze]/J; RRID: IMSR_JAX:007909) to obtain red fluorescent labeling of *vGlut2*-expressing cells. Animals were then bred at Northeast Ohio Medical University maintaining a 12/12 h light cycle with ad libitum food and water. Animals of both sexes were used. At the time of electrophysiological recordings mice (33) were aged 21–44 days, 27 ± 0.59 avg and gerbils (23) were 21–36 days, 28 ± 0.67 avg.

**Electrophysiology and intrinsic membrane properties**. Animals were decapitated under isoflurane anesthesia and the brain quickly removed in oxygenated cutting solution containing in mM: 135 N-methyl-D-glucamine (NMDG, Sigma), 1.25 KCl, 1.25 KH$_2$PO$_4$, 0.5 CaCl$_2$, 2.5 MgCl$_2$, pH 7.35 with HCl, ~310 mmol/kg. The brainstem was isolated, embedded in 1% agarose and sliced coronally 180–200 μm thick using a vibrating microtome (7000 smz2, Campden, UK) at room temperature. Slices were transferred to a recovery solution containing in mM: 110 NaCl, 2.5 KCl, 1.5 CaCl$_2$, 1.5 MgCl$_2$, 25 NaHCO$_3$, 1.25 NaH$_2$PO$_4$, 12 dextrose, 5 N-Acetyl-L-cysteine, 5 Na-ascorbate, 3 Na-pyruvate, 2 thiourea, pH 7.35 with NaOH, continuously bubbled with 5% carbogen, ~295 mmol/kg at 35 °C for 30 min. and then maintained at room temperature until being transferred to the recording stage. Recordings were made in oxygenated artificial cerebrospinal fluid (ACSF) containing in mM: 120 NaCl, 2.5 KCl, 1.5 CaCl$_2$, 1.5 MgCl$_2$, 25 NaHCO$_3$, 1.25 NaH$_2$PO$_4$, 12 dextrose, pH 7.35 with NaOH, ~295 mmol/kg at 35 ± 0.5 °C maintained using an in-line heating system.

Neurons were targeted mostly using infrared Dodt contrast combined with two-photon fluorescence (Hyperscope, 20X NA 1 objective, Scientifica) or differential interference contrast microscopy combined with widefield fluorescence (Axioskop 2 FS Plus, ×40 NA 0.8 objective, Zeiss). LSO PNs were distinguished from olivocochlear cells based on size, shape, and membrane properties in current-clamp mode[24,35]. Specifically, PNs are larger (approximately double soma size), fusiform shape (Figs. 1b and 5a–d) and exhibit more hyperpolarized resting potentials compared to olivocochlear cells. Olivocochlear cells also exhibit a characteristic

delay-to-firing response not observed in PNs and lack the prominent $I_h$/HCN channel currents in response to hyperpolarization that are observed in PNs.

Whole-cell patch-clamp recordings were made using Dual IPA (Sutter Instruments, CA) or EPC-10 USB (HEKA, Germany) amplifiers with integrated digitizers in current-clamp mode using thick-walled borosilicate patch pipettes (4–6 MΩ, Sutter) filled with K-gluconate internal containing in mM: 115 K-gluconate, 8 KCl, 0.5 EGTA, 10 HEPES, 10 $Na_2$ phosphocreatine, ~22 sucrose, 4 Mg-ATP, 0.3 Na-GTP, 0.1% (2.68 mM) biocytin, 0.05 Alexa 488 hydrazide, pH 7.30 with KOH, ~297 mmol/kg, ECl = −74 mV at 35 °C. Data were low-pass filtered at 5 kHz or 2.9 kHz, digitized at 50 kHz, and acquired to computer using custom macros for IgorPro (Wavemetrics, OR) or PatchMaster Next (HEKA). Series resistances were <25 MΩ (14 ± 0.4 MΩ). A measured liquid junction potential of −11 mV was corrected, however, the theoretical value was −15 mV[69]. Resting membrane potential was recorded immediately after break-in. Intrinsic membrane response properties were measured at −66 mV maintained with small DC current injections when necessary. Rheobase was defined as the first current level at which an AP occurred measured at a resolution of 10 pA step size for 200 ms. AP parameters (Fig. 2a) were measured at rheobase.

Statistical comparisons were made using Prism (GraphPad, San Diego, CA). Comparisons between transmitter types alone were made using two-tailed, unpaired $t$-test. Comparisons across transmitter types broken up into AP firing types were made using two-way ANOVA and significance stated on graphs (*s) from multiple comparisons with Tukey corrections. Significance was assessed using an alpha level of 0.05.

**Spiking analysis**. Spike count and timing analyses were carried out in MATLAB (Mathworks, Natick, MA) on 200 ms current steps at 100–150 pA intervals transformed into differential voltage traces to better distinguish spikes/APs from spikelets. Differential voltage traces were low-pass filtered using a 4th order Butterworth filter with a scale-free upper frequency cutoff of 7.5 kHz. A criterion of 62.5 mV/ms was chosen using visual analysis of AP threshold "kink" to identify spike events and extract first spike latency, initial firing rate, and inter-spike interval (ISI). Dynamic range (DR) levels were computed from current steps, ranging from rheobase at a DR level of 0 and an overload current where firing rate decreased or became unstable at a DR level of 1:

$$\text{DR level} = \frac{\text{current step amplitude} - \text{rheobase}}{\text{overload current} - \text{rheobase}} \quad (1)$$

As the number of DR levels varied between neurons, intermediate response parameters were obtained from interpolation at an incremental DR level of 0.001. ISI adaptation was computed as the ratio of the average of the last 3 ISIs to the average of the first 3 ISIs. ISI ratios were interpolated across DR levels, and the values were binned at 0.2-DR resolution.

To analyze cell-type differences in spiking parameters, a linear mixed model (LMM) using the fitlme() MATLAB function was run on non-interpolated response data to account for unequal number of neurons under firing type, neurotransmitter type, and missing values for some current amplitudes in absolute or DR scale. The fixed effects factors in the model were intercept and the main effects. The main effects were DR level or absolute current, firing type, and neurotransmitter type. All pairs of interactions between the main effects were considered. Random effects of neuron ID on DR level and on intercept in the model accounted for idiosyncratic variability in the DR levels and intercept across neurons. Model coefficients were considered significant at an alpha level of 0.05. The $p$ values were reported from model $t$-tests of coefficients for main effects of firing type and neurotransmitter type, significant interactions between current level and firing type (Figs. 3d and 4a) or between current level and neurotransmitter type (Fig. 3l). For a significant interaction between neurotransmitter type and firing type, $p$ values were either reported from model $t$-test (Fig. 3c) or additionally reported from post hoc model $F$-tests between neurotransmitter types within each firing type (Fig. 2d).

Time coding analyses were restricted to the first 20 ms. Pulse sequences of varying DR levels and time durations were used. Spike times were interpolated across DR levels. For each pulse DR level, trial spike trains were drawn for 10 adjacent DR levels. The same sequence of pulse DR levels was used across all neurons for response comparisons across firing types. Pulse sequences of mean DR levels 0.05, 0.5, and 0.95 were used to mimic stimulation at activation threshold, mid-range, and saturation while avoiding floor and ceiling effects from sampling at the extremes.

Temporal phase-locking was assessed using cycle-by-cycle vector strength (VScc)[70,71] computed from the 10 trials for each pulse DR level in the sequence where $n$ is the number of spikes over the 10 trials at a given DR level in the sequence, $\theta_i$ is the phase of $i$th spike in radians, $t_i$ is the time of the $i$th spike in ms, and $T$ is the temporal width of the pulse sequence in ms:

$$\text{VScc} = \frac{\sqrt{(\sum_{i=1}^{n} \cos\theta_i)^2 + (\sum_{i=1}^{n} \sin\theta_i)^2}}{n}, \theta_i = 2\pi \frac{t_i \text{ modulo } T}{T} \quad (2)$$

Pulse-wise VScc was set to zero if none of the trials had a spike for a given pulse DR level and temporal width. The overall strength or mean VScc at the mean DR level was the average of all pulse-wise VScc values. Then, population-level mean

and variance of mean VScc were determined for the temporal widths between 0.5 and 20 ms.

DR level coding strength was quantified by population mutual information (MI) between population mean spike count ($R$) for various temporal periods from current onset and DR level ($L$)[72], given below, where $P(.|.)$ is the conditional probability and $P(.)$ the marginal probability:

$$\text{Population MI}(R;L) = \sum_{R} \sum_{L} P(R|L) \times P(L) \times \log_2 \frac{P(R|L)}{P(R)}, P(R) = \sum_{L} P(R|L) \times P(L) \quad (3)$$

As DR level is taken to be uniformly random, $P(L)$ is simply the reciprocal of total number of interpolated DR levels. Here, $P(R|L)$ was assumed to be Poisson distributed and depended only on mean spike counts at the DR level $L$. A spike count histogram-based $P(R|L)$ also yielded similar results. Population MI was additionally normalized such that a value of 0 is absent level coding and a value of 1 indicates that every spike count is associated with a unique DR level, where the denominator is total entropy in bits, and $N$ is the number of interpolated DR levels:

$$\text{Normalized population MI}(R;L) = \frac{\text{Population MI}(R;L)}{\log_2 N} \quad (4)$$

**Morphological analysis**. Our internal solution contained 0.05 mM Alexa 488-hydrazide which allowed us to image the majority of cells using two-photon microscopy immediately after electrophysiological recordings and eliminate shrinkage effects associated with fixed tissue imaging. Images were acquired using a Scientifica Hyperscope two-photon microscope at 765 nm with resonance scans using a ×20 NA1 objective. Morphology was examined in 70 neurons from 1–2 µm z-stacks of images with 512 × 512 pixel resolution.

Arbor reconstruction was done using Simple Neurite Tracer[73] in Fiji/ImageJ[74] and analyzed alongside somata and manually drawn LSO boundaries using custom MATLAB scripts (Fig. 5a–d).

Dendritic eccentricity ($E$) was calculated from the maximum angle between tip-to-soma-center vectors ($\theta_{max}$) with a centered soma configuration set to 0 and eccentric soma configuration having a value of 1 on the extremes:

$$E = \frac{(1 + \cos(\theta_{max}))}{2} \quad (5)$$

Inclusion of very short dendrites that pointed in the opposite direction decreased eccentricity of cells that largely had an eccentric morphology. To exclude such dendrites, a dendritic tip was only included if its distance from the soma center was at least 0.4 times the maximum dendritic tip-to-soma-center distance. For neurons with eccentricity of less than 0.9 and dendritic extension from soma of at least 55 µm, dendritic orientation axis was computed in the coronal plane as the line connecting the medians of the tips on either side of the soma along the $x$-axis (mediolateral) using a less conservative dendritic tip inclusion criterion of 0.2 times the maximum dendritic tip-to-soma-center. However, if the tips were spread more along $y$-axis (dorsoventral), then medians were calculated on either side of the soma in the y-direction. A presumptive cross-frequency spread was computed as the maximum dendritic spread about the soma center along an axis orthogonal to the dendritic orientation axis.

Coronal (xy) spread of dendritic tips was measured as the maximum distance between the dendritic tips in the image plane and the z-spread as the maximum and minimum $z$ values for the dendritic tips.

Statistical comparisons for morphology were made using Prism. Comparisons between transmitter types alone were made using two-tailed, unpaired $t$-test. Comparisons across transmitter types broken up into AP firing types were made using two-way ANOVA and significance stated on graphs (*s) from multiple comparisons with Tukey corrections. Significance was assessed using an alpha level of 0.05. Cell location in LSO and tonotopic analyses were performed with chi-squared test.

**Statistics and reproducibility**. The quantitative cellular data presented is expressed as mean ± standard error of the mean (SEM) where $n$ = cells. SEM was used as a measure of reproducibility. Sample size was not predetermined but based on experience and number of cells in each group. Significance was assessed using an alpha level of 0.05. Comparisons between two groups based on gene/neurotransmitter expression were made using $t$-tests. Additional comparisons were made using ANOVA at additional grouping levels based on response pattern, sex, and location within the nucleus. Where interactions were found, post hoc analysis using Tukey corrected multiple comparisons were reported. Linear mixed models were used where there were repeated measures of current amplitudes with missing values due to sampling differences. An exploratory approach using principal component analysis was taken to assess the validity of analysis groupings. Categorical data presented was assessed using chi-squared test.

**Reporting summary**. Further information on research design is available in the Nature Portfolio Reporting Summary linked to this article.

## Data availability
Numerical source data for the study are available in Supplementary Data 1.

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

## Acknowledgements

This work was supported by the United States National Institutes of Health, Institute on Deafness and Other Communication Disorders (NIDCD) R21 DC017819 Winters. We thank Dr. Joshua H. Goldwyn for valuable input on data analysis and the manuscript.

## Author contributions

H.H. conducted experiments, performed analysis, and wrote the manuscript. B.D.W. conducted experiments, performed analysis, designed research, and wrote the manuscript.

## Competing interests

The authors declare no competing interests.
