## [Peer Review File · Communications Biology]

Reviewers' comments:

Reviewer #1 (Remarks to the Author):

Comments

Re: VGlut+ and VGlut- cells. Have data been published on the mouse they created? It is not unusual for knock-ins or knock-outs not to be the same as when immunostaining for the molecule knocked-in or knocked-out (Blusztajn & Rinnofner, *Frontiers in Synaptic Neuroscience* 8. doi:10.3389/fnsyn.2016.00006). Some control data on this mouse is essential.

Line 101: how do the authors distinguish high from medium or low glycine immunoreactivity? Does VGlut2 negativity mean GlyT2 positivity? I did not see anything specific to this issue. When performing immunostaining, the nature of the fixative is important. The online Sicgen product page (the authors source of antibodies) uses 4% paraformaldehyde as a fixative for their immuno figure. For this paper, the authors fixed their tissue with glutaraldehyde, a strong fixative known to alter antigenicity. Are the results in the immunostaining misleading because of fixation artifact?

The physiological characterization of LSO PNs is thorough--starting with line 138, the authors discuss both mouse and gerbil but are often not clear with identifying the species that contribute data.

Maybe put the gerbil data in as supplementary material--the gerbil data do not add to the story.

Does inhibitory mean vGlut2- or GlyT2+??

Paragraph beginning on line 405+: confusing to say that inhibitory and excitatory LSO PNS were similar but transmitter types did differ. What do they mean? Do soma v dendritic features differ with respect to transmitter? Do the cumulative data result in there being just 2 groups of LSO PNs--excitatory and inhibitory (Supplemental Data 4)? The authors imply that inhibitory LSO PNs project ipsilaterally, whereas excitatory LSO PNs project contralaterally. Is this the segregation part of sound localization strategy?

lines 481-: There isn't really a disagreement on firing types; there is just differences in the techniques used to derive firing types. That is, are data collected from an intact animal or data from cells/slice a dish? The dish is totally abnormal and only an approximation of what goes on in the intact brain. There is also the issue of age. Dish recordings are typically made from young animals. The age of mice in this report is just younger than the average range of 6-20 weeks reported in a large survey of rodent researchers (Jackson et al., *Lab Anim.* 51(2)160-168, 2017). Is some of the variability in the data due to maturational changes?

line 524-525: the argument about marginal cells is unproductive; the authors are arguing about a name. there is no evidence that a "marginal cell" actually exists other than it sits on the margin of the nucleus. Any entity on any border might appear different and could have unique properties but that needs to be proven. There are other cells in the LSO that lie along the border that are not called marginal cells.

line 526: This sentence is incorrect. Of course synapse coverage on the cell body has functional significance. The paragraph needs to be rewritten.

Figure 6 The use of the term "tuned" is not well defined. It is a generic term used to describe any of a variety of receptive field parameters to which a neuron might show a response preference. Cells get "tuned" with respect to tone frequency, interval between two stimuli, color, visual location, etc.); the authors need to be more explicit in their use of "tuning."

Lines 566-: The authors statement that there is "clearer picture of cellular diversity" is not obvious from their data. There is diversity of many observations but that doesn't create a clear picture.

•VGlut2+ cells: no different in somatic volume between + and - cells.

- VGlut2- cells:
- GlyT2+ Immuno--are these 100% the same as the VGlut-cells? We only know they don't overlap with VGlut2+ cells
- Onset/bursting cells
- Multi-spiking cells
- VGlut2+ contralateral projections; longer dendrites w more branches;
- VGlut2- ipsilateral projections?
- VGlut+/- cells were fusiform/bipolar recorded from primarily in the middle of the LSO; and they have same firing types no influence on transmitter type
- What does z-spread mean? It will be limited by the thickness of the tissue section and the degree of shrinkage. It is not a biological feature.
- Photos of cells (Fig. 5A) too small to gain real appreciation of the structure; they should show off their cells because they are rare. Maybe compare to anatomy figures shown in the Franken et al. 2018 publication.
- There are many physiological parameters--tau, threshold, Vm, Sag, etc (Figs 1 and 2). What is the significance? These are more intrinsic values than information related to segregation of processing lines.

There are a lot of facts presented in this manuscript but the authors have not convincingly organized the data into a coherent whole. They are not explicit in their conclusion and they fall short of the goal stated in the title.

Reviewer #3 (Remarks to the Author):

The manuscript "Cellular Diversity in the Lateral Superior Olive to Support Multiple Sound Localization Strategies and Segregate Information in Higher Processing Centers" by Haragopal and Winters provides a carefully executed survey of identified classes of LSO neurons in the mouse and corresponding data from the gerbil. This study is the first to provide a thorough analysis of intrinsic membrane physiology in LSO neurons, and reveals some response characteristics that are aligned with the LSO's known functions in sound localization. The addition of anatomical data is welcome and provides some additional context. The diversity of response types reported here creates a challenge for interpretation, but the authors have at the very least proposed a number of testable hypotheses for future work based on these results. Overall the science and the presentation are very good to excellent, but a number of minor changes would greatly enhance the efficiency of presentation and readability. I share some minor critiques in the hope that these suggestions would improve the manuscript.

-Title- It seems the title is lacking a verb and without one it is awkward to read

Results:

-The results section is somewhat difficult to read as it is heavily biased toward a thorough but iterative list of statistical comparisons without context. I think for a non-specialist readership the manuscript could be improved by simply adding section headings that point to major findings and brief single sentence summaries added to the end of major sections to move the narrative along and provide transitions. Some authors prefer a sparse, data-only results section, but in this case you are risking losing your audience's attention.

-AP half width is presented at line 195 and then again at 224 creating a redundancy, I suggest merging at one point or the other

-The results section is replete with instances of "we wanted to," eg 237-254-272 in my opinion the results are about what was accomplished and not what the authors 'wanted' to accomplish. These transitions would be better represented by "Next we investigated..." or similar.

-line 248: with regards to--should be 'with regard to'

-line 265: where spiking decreased- is this an intrinsic nonmonotonicity? please explain, it might be useful to describe the degree and manner of spike count decline with strong depolarization for auditory modelers.

-line 238 "We divided the LSO into 3 equal parts along the midline" Do you mean parallel to the midline? 'along the midline' is a bit unclear- referencing one of the illustration in a slightly modified existing figure would be helpful- especially since the dendritic projections are often described with reference to tonotopic lamina. It is a weakness to expect the audience to have a well-informed mental model of LSO frequency representation and circuitry in these two species.

-on this note, I was expecting to see tonotopy appear earlier in the manuscript especially since the gerbil data may be fundamentally different in this regard if CF is a factor of variance. It would be nice to simply mention this analysis was included earlier in the manuscript, perhaps at the end of the introduction or in the initial findings of the results.

Response to reviewers' comments: COMMSBIO-22-4193-T, Winters

Reviewer #1 (Remarks to the Author):

Comments

Re: VGlut+ and VGlut- cells. Have data been published on the mouse they created? It is not
unusual for knock-ins or knock-outs not to be the same as when immunostaining for the
molecule knocked-in or knocked-out (Blusztajn & Rinnofner, *Frontiers in Synaptic Neuroscience*
8. doi:10.3389/fnsyn.2016.00006). Some control data on this mouse is essential.

The reviewer raises a common concern for using transgenic mice such as those created using
bacterial artificial chromosome (BAC) system which is the subject of the provided reference
(Blusztajn, 2016). In such transgenic mice the gene reporter sequence is randomly inserted into
the genome. This leads to variability in expression due to a variety of factors such as local
chromatin structure at the insertion site that are unknown. The mouse line used in our study is
not subject to these problems because it is a knock-in model in which the reporter sequence is
placed into the gene of interest through targeted insertion. This ties reporter expression to
expression of the gene of interest. Additionally, the reporter protein is not attached to the gene
product of interest (vGlut2 in our case) so there are no issues with affecting protein function.
The mouse line used in our study is obtained from Jackson Labs (Strain #:028863) and is an
extremely well-validated model organism. The original article from the Lowell lab (Vong, 2011)
characterized reporter expression throughout the brain. This mouse line has also been
specifically validated in brainstem tissue in recent articles (Pauli, 2022; Doykos, 2020) one of
which was in a premier neuroanatomy journal (Doykos, 2020). In our supplementary figure 1 we
demonstrate that vGlut2 positive cells from this mouse line do not exhibit high intracellular
glycine associated with inhibitory cells in our region of interest suggesting that there is no
overlap between cell types as was previously demonstrated with in situ hybridization (Mellott,
2021; Ito and Oliver, 2011). Thus, we feel that this mouse line does not require further
validation.

Line 101: how do the authors distinguish high from medium or low glycine immunoreactivity?
Glycine immunoreactivity was assessed visually relative to other cells in the same micrograph
as described in Fig. S1.

Does VGlut2 negativity mean GlyT2 positivity? I did not see anything specific to this issue.
Not explicitly, however, Fig. S2 demonstrates that putative GlyT2 cells are not vGlut2 positive.
Additionally, only glycinergic and glutamatergic LSO PNs have been described and vGlut2 is
obligatory for glutamatergic LSO PNs and vGlut2 does not overlap with inhibitory markers GlyT2
or VIAAT as described in the first results paragraph.

When performing immunostaining, the nature of the fixative is important. The online Sicgen
product page (the authors source of antibodies) uses 4% paraformaldehyde as a fixative for
their immuno figure. For this paper, the authors fixed their tissue with glutaraldehyde, a strong
fixative known to alter antigenicity. Are the results in the immunostaining misleading because of
fixation artifact?

The glycine antibody we used was intentionally created using a glutaraldehyde-treated antigen
so the tissue must be fixed in glutaraldehyde (Fig. S2 methods). The Sicgen antibody was used
to recover/amplify the tdTomato signal in vGlut2 expressing cells after the fixation/quenching
process. No issues with penetration or unusual background were observed for the tdT antibody.
Additionally, the demonstrated anticorrelation between the two antibodies suggests that tdT
antigenicity was not perturbed.

The physiological characterization of LSO PNs is thorough--starting with line 138, the authors
discuss both mouse and gerbil but are often not clear with identifying the species that contribute
data.

We have added numerous additional clarifications to the manuscript where mouse tissue is
being described as suggested. Additionally, since excitatory/inhibitory type cannot be
determined in gerbils, the species is implied when discussing transmitter groups.

Maybe put the gerbil data in as supplementary material--the gerbil data do not add to the story.
The gerbil data is provided alongside the mouse data so that it can be visually compared. This
information is important for several reasons. The most recent in vivo characterizations of LSO
cell types (Franken, 2018, 2021) were completed in gerbil tissue. This work is somewhat
controversial as it suggest that the traditional thinking on LSO firing types, chopper cells
encoding ILDs, is wrong. Since inhibitory/excitatory cell types have not been examined and we
found different results from these authors regarding firing type distribution using mice, it is
important to show that we also find these feature in gerbils. Additionally, the comparative

approach between gerbils that hear low frequencies and mice that do not is relevant to research
aimed at understanding how the LSO system works as a whole and in humans.

Does inhibitory mean vGlut2- or GlyT2+??

Inhibitory is defined early in the results section (~line 101) as vGlut2 negative.

Paragraph beginning on line 405+: confusing to say that inhibitory and excitatory LSO PNS
were similar but transmitter types did differ. What do they mean? Do soma v dendritic features
differ with respect to transmitter?

These results are describing morphological features. We state that the transmitter types were
similar in some respects (soma volume, max extension, dendrite dia) but different in others
(total length, number of branch points, number of primary dendrites).

Do the cumulative data result in there being just 2 groups of LSO PNs--excitatory and inhibitory
(Supplemental Data 4)?

Figure S4 is looking at electrophysiological parameters using principal component and cluster
analysis. These demonstrate in an unbiased manner that it is likely that there are two groups
and that transmitter type explains most of the variability. This does not mean that firing type
does not differentiate groups, as several parameters were firing-type-dependent.

The authors imply that inhibitory LSO PNs project ipsilaterally, whereas excitatory LSO PNs
project contralaterally. Is this the segregation part of sound localization strategy?

The segregation of LSO PN projections to IC have been explored in several species and is well
established (see discussion first paragraph). The excitatory projection differs between low-
frequency and non-low-frequency animals suggesting that this may indeed be involved in sound
localization however we do not address this in the present study.

lines 481-: There isn't really a disagreement on firing types; there is just differences in the
techniques used to derive firing types. That is, are data collected from an intact animal or data
from cells/slice a dish? The dish is totally abnormal and only an approximation of what goes on
in the intact brain.

The firing type of LSO PNs is not settled. As noted in the manuscript, there are difference
between in vivo reports with some authors finding all or mostly chopper/multi-spiking cells and
others inferring that most LSO PNs are true onset firing. There is also disagreement between ex

vivo/in vitro reports. Thus, the disagreement is not as simple as a difference between in vivo
and in vitro (dish) experiments. The opinion that “dish” experiments are abnormal or not
representative is not well-founded, especially for acute brain slices, as opposed to cell cultures,
in which the support mechanisms and synaptic circuits are largely intact. In the current case, our
ex vivo findings are similar to the most recent in vivo reports (Franken, 2018, 2021) that also
used whole-cell technique in that we both find onset and multi-spiking firing types. Additionally,
these authors showed that whole-cell current injections closely mimic in vivo responses to tone
bursts. We are expanding on this knowledge using an ex vivo approach that allows for a much
larger data set and further dissection of LSO cell types.

There is also the issue of age. Dish recordings are typically made from young animals. The age
of mice in this report is just younger than the average range of 6-20 weeks reported in a large
survey of rodent researchers (Jackson et al., Lab Anim. 51(2)160-168, 2017). Is some of the
variability in the data due to maturational changes?

The age of animals for the current study was on average P27 and older than the majority of
prior ex vivo studies of LSO PN intrinsic membrane properties. The auditory system in rodents
of this age has reached mature thresholds and animals are well-capable of sound localization.
Indeed, at this age wild mice would be foraging on their own and would need an effective sound
localization system for survival. Regarding variability, we do not attempt to rule out age-related
variability, however, the substantial differences observed for most parameters exhibited low
variability as evidenced by very small p values.

line 524-525: the argument about marginal cells is unproductive; the authors are arguing about
a name. there is no evidence that a "marginal cell" actually exists other than it sits on the margin
of the nucleus. Any entity on any border might appear different and could have unique
properties but that needs to be proven. There are other cells in the LSO that lie along the border
that are not called marginal cells.

We agree with the reviewer's viewpoint and indeed our data and our discussion of it suggests
that there is not a distinct set of marginal cells with multi-spiking response pattern. However, a
discussion regarding marginal cells and the relationship between our data and previous work is
required because it is a key claim made by prior in vivo reports (Franken, 2018, 2021) which
suggested that multi-spiking LSO PNs are a small population and limited to marginal cells. Such
a configuration would be important to understand as it would impact the functional integrative
capability of the LSO. Our findings directly refute this claim and help clarify the LSO circuit.

line 526: This sentence is incorrect. Of course synapse coverage on the cell body has functional
significance. The paragraph needs to be rewritten.

The sentence has been rewritten to clarify that we are discussing functional significance for
sound localization in this specific circuit not in general stating: "It is not known whether the
degree of somatic synapse coverage has functional significance for sound localization or
association with projection pattern in the LSO." If the reviewer knows of some form of functional
relevance for somatic synapse coverage for the LSO circuit a reference would be much
appreciated.

Figure 6 The use of the term "tuned" is not well defined. It is a generic term used to describe
any of a variety of receptive field parameters to which a neuron might show a response
preference. Cells get "tuned" with respect to tone frequency, interval between two stimuli, color,
visual location, etc.); the authors need to be more explicit in their use of "tuning."

The text has been revised to clarify that the tuning referred to is for location of sound source
stating "illustration of 2 cells similarly tuned for a particular location with the sound source near
their transition location."

Lines 566-: The authors statement that there is "clearer picture of cellular diversity" is not
obvious from their data. There is diversity of many observations but that doesn't create a clear
picture.

The statement in question reads "Although questions remain, together these data bring a
clearer picture of cellular diversity in the LSO..." We feel that this statement does not inflate the
importance of our work. We do not claim to have clarified everything about the LSO. Simply that
we have contributed to the knowledge about LSO cell types making it clearer. Indeed, this is not
unfounded given that LSO PNs differ in their transmitter systems and projection patterns, thus
potentially play different roles in the circuit, but have always been lumped together and have
never been characterized separately!

•VGlut2+ cells: no different in somatic volume between + and - cells.

This seems to be a correct statement, but we are unsure what is being asked.

•VGlut2- cells:

•GlyT2+ Immuno--are these 100% the same as the VGlut-cells? We only know they don't
overlap with VGlut2+ cells

This seems to be a correct statement, but we are unsure what is being asked.

•Onset/bursting cells

•Multi-spiking cells

•VGlut2+ contralateral projections; longer dendrites w more branches;

This seems to be a correct statement, but we are unsure what is being asked.

•VGlut2- ipsilateral projections?

This seems to be a correct statement according to prior reports, but we are unsure what is being
asked.

•VGlut+/- cells were fusiform/bipolar recorded from primarily in the middle of the LSO; and they
have same firing types no influence on transmitter type

Cells were recorded throughout the body of the LSO. There were both firing types in both
transmitter types, however, the ratio of firing types differed between transmitter types.

•What does z-spread mean? It will be limited by the thickness of the tissue section and the
degree of shrinkage. It is not a biological feature.

The z-axis was measured in live slices ~200 microns thick at the time of recording using two-
photon microscopy to eliminate shrinkage/fixation as a factor and virtually all cells were
completely contained in the z/depth imaged. The manuscript has been modified to make this
point clear.

•Photos of cells (Fig. 5A) too small to gain real appreciation of the structure; they should show
off their cells because they are rare. Maybe compare to anatomy figures shown in the Franken
et al. 2018 publication.

The figure provides representative images for the ~70 dendritic reconstructions we obtained. It
is not clear what is meant by rare. There have been several prior reports by Sanes and others of
the dendritic anatomy of LSO PNs. We are however the first to compare inhibitory and

excitatory LSO PNs. Franken had few precious and important recordings as they were the first
to perform difficult whole-cell recording in LSO and thus showed all of the reconstructions.

•There are many physiological parameters--tau, threshold, Vm, Sag, etc (Figs 1 and 2). What is
the significance? These are more intrinsic values than information related to segregation of
processing lines.

Since these cell types had not been separated before, we provide a thorough characterization of
their intrinsic membrane properties. Many of these parameters were very significantly different
between groups. Information segregation is indeed an important finding of the study as not only
are there labeled lines in upstream processing centers by transmitter system, and by projection
side as show by prior reports, but here we show that there are different activation thresholds as
well. Additionally, the potential differences in integrative properties between groups due to
factors such as dendritic arbor size and ratio of firing types may provide differentially extracted
information to these segregated lines.

There are a lot of facts presented in this manuscript but the authors have not convincingly
organized the data into a coherent whole. They are not explicit in their conclusion and they fall
short of the goal stated in the title.

We would refute this statement by saying that indeed lots of data ("facts") needed to be
presented to since these cell types had not been explored before. Much of this information will
be of value for future modelling or in vivo studies and therefore is presented completely.

We are unsure how our conclusions are not explicit enough. We do not wish to overstate
the functional implication of our findings as we have not yet tested them using modelling or in
vivo recordings, though these are future directions. Nonetheless, we think that the dual role
model presented is a novel platform from which to test hypotheses of LSO function.

We are unsure what the reviewer felt the goal stated by the title is, however, we
observed cellular diversity in the LSO PN population that would allow the LSO to perform both
ITD and ILD functions and potential new ways for information to be segregated in higher
processing centers such as activation threshold. Therefore, we believe the title is apt and not
overstated.

Reviewer #3 (Remarks to the Author):

The manuscript "Cellular Diversity in the Lateral Superior Olive to Support Multiple Sound
Localization Strategies and Segregate Information in Higher Processing Centers" by Haragopal
and Winters provides a carefully executed survey of identified classes of LSO neurons in the
mouse and corresponding data from the gerbil. This study is the first to provide a thorough
analysis of intrinsic membrane physiology in LSO neurons, and reveals some response
characteristics that are aligned with the LSO's known functions in sound localization. The
addition of anatomical data is welcome and provides some additional context. The diversity of
response types reported here creates a challenge for interpretation, but the authors have at the
very least proposed a number of testable hypotheses for future work based on these results.
Overall the science and the presentation are very good to excellent, but a number of minor
changes would greatly enhance the efficiency of presentation and readability. I share some
minor critiques in the hope that these suggestions would improve the manuscript.

We thank the reviewer for their efforts and constructive suggestions.

-Title- It seems the title is lacking a verb and without one it is awkward to read

We tend to agree with this and went back and forth some with it ourselves. We must be careful
not to overstate or get too long. One option is:

"Investigation of transmitter types in the lateral superior olive reveals cellular diversity to support
multiple sound localization strategies and segregate information in higher processing centers"

or shorten it to

"Investigation of transmitter types in the lateral superior olive reveals cellular diversity to support
multiple sound localization strategies and segregate information"

Results:

-The results section is somewhat difficult to read as it is heavily biased toward a thorough but
iterative list of statistical comparisons without context. I think for a non-specialist readership the
manuscript could be improved by simply adding section headings that point to major findings
and brief single sentence summaries added to the end of major sections to move the narrative

along and provide transitions. Some authors prefer a sparse, data-only results section, but in
this case you are risking losing your audience's attention.

We have added additional subheadings that are descriptive to the results section and some
intermediate conclusions are included in the results for subjects that are not discussed in the
discussion section.

-AP half width is presented at line 195 and then again at 224 creating a redundancy, I suggest
merging at one point or the other

We performed a couple of levels of analysis to assess different hypotheses, so all of the action
potential parameters, not just half width, appear twice. Once to compare just between
transmitter types as a whole and once to test for effects of firing type as well as transmitter type.
If we were to group by property measured, it would require multiple explanations of what was
being compare, thus this would not be our choice for organization.

-The results section is replete with instances of "we wanted to," eg 237-254-272 in my opinion
the results are about what was accomplished and not what the authors 'wanted' to accomplish.
These transitions would be better represented by "Next we investigated..." or similar.

All instances of "wanted" have been modified as suggested.

-line 248: with regards to--should be 'with regard to'

All instances of "regards to" have been modified as suggested.

-line 265: where spiking decreased- is this an intrinsic nonmonotonicity? please explain, it might
be useful to describe the degree and manner of spike count decline with strong depolarization
for auditory modelers.

We are not sure what the ionic basis of nonmonotonicity might be, but it is an interesting idea.
When injecting currents, overloading tends to occur as the voltage gated sodium channels are
no longer able to recover from inactivation. This usually manifests as a decreasing spike
amplitudes into petering out of spiking all together toward the end of the sweep. A description of
this behavior has been added as suggested.

-line 238 "We divided the LSO into 3 equal parts along the midline" Do you mean parallel to the
midline? 'along the midline' is a bit unclear- referencing one of the illustration in a slightly
modified existing figure would be helpful- especially since the dendritic projections are often

described with reference to tonotopic lamina. It is a weakness to expect the audience to have a
well-informed mental model of LSO frequency representation and circuitry in these two species.
The statement has been modified to clarify how the LSO was divided and reference our prior
work where this is illustrated stating: "We divided the LSO by bisecting the outline of the LSO
from tip to tip to create a curving midline then dividing the nucleus into 3 equal parts
perpendicular to midline to crudely represent high (medial), middle, and low frequency (lateral)
regions (Mellott et al., 2021)."

-on this note, I was expecting to see tonotopy appear earlier in the manuscript especially since
the gerbil data may be fundamentally different in this regard if CF is a factor of variance. It would
be nice to simply mention this analysis was included earlier in the manuscript, perhaps at the
end of the introduction or in the initial findings of the results.

Mention of tonotopic analysis has been added to the end of the introduction as suggested.

REVIEWERS' COMMENTS:

Reviewer #3 (Remarks to the Author):

I have reviewed the response of the authors and the changes to the manuscript. I am satisfied that they have addressed my concerns. I expect that this paper will be an important contribution to the LSO and sound localization literature. I have no further concerns.